# CONTINUUM TRANSFORMERS PERFORM IN-CONTEXT LEARNING BY OPERATOR GRADIENT DESCENT

**Abhiti Mishra**[*]
Department of Statistics
University of Michigan
Ann Arbor, MI 48104
abhiti@umich.edu

**Yash Patel**[*]
Department of Statistics
University of Michigan
Ann Arbor, MI 48104
yppatel@umich.edu

**Ambuj Tewari**
Department of Statistics
University of Michigan
Ann Arbor, MI 48104
tewaria@umich.edu

## ABSTRACT

Transformers robustly exhibit the ability to perform in-context learning, whereby their predictive accuracy on a task can increase not by parameter updates but merely with the placement of training samples in their context windows. Recent works have shown that transformers achieve this by implementing gradient descent in their forward passes. Such results, however, are restricted to standard transformer architectures, which handle finite-dimensional inputs. In the space of PDE surrogate modeling, a generalization of transformers to handle infinite-dimensional function inputs, known as "continuum transformers," has been proposed and similarly observed to exhibit in-context learning. Despite impressive empirical performance, such in-context learning has yet to be theoretically characterized. We herein demonstrate that continuum transformers perform in-context operator learning by performing gradient descent in an operator RKHS. We demonstrate this using novel proof strategies that leverage a generalized representer theorem for Hilbert spaces and gradient flows over the space of functionals on a Hilbert space. We further show the operator learned in context is the Bayes Optimal Predictor in the infinite depth limit of the transformer. We then provide empirical validations of this result and demonstrate that the parameters under which such gradient descent is performed are recovered through pre-training.

## 1 INTRODUCTION

LLMs, and transformers more broadly, have demonstrated a remarkable ability to perform in-context learning, in which predictive performance improves without the need for parameter updates, merely by providing training samples in the LLM context window Minaee et al. (2024); Dong et al. (2022). With workflows increasingly relying on such fine-tuning in place of traditional weight updates due to its computational efficiency, much interest has gone into providing a theoretical explanation of this phenomenon Akyürek et al. (2022); Garg et al. (2022); Dai et al. (2022). Such works have demonstrated that, with particular choices of transformer parameters, an inference pass through such models is equivalent to taking steps of gradient descent for the in-context learning task.

While LLMs were the first setting in which in-context learning abilities of transformers were exploited, interest is increasing in its broader application to other sequence prediction tasks. In particular, a seemingly orthogonal subfield of machine learning is that of accelerating the solution of partial differential equations (PDEs) Li et al. (2020a); Du et al. (2023); Liu et al. (2023); Jafarzadeh et al. (2024); Oommen et al. (2024); You et al. (2022). In this setting, the sequence is no longer of finite-dimensional vectors, but instead of infinite-dimensional functions. Nonetheless, transformers have been adapted to this setting, with an architecture known as "continuum transformers" Calvello et al. (2024). More surprising still is that such transformers continue to exhibit in-context learning, whereby related PDEs can be efficiently solved with the similar placement of solution pairs in the continuum transformer context window Cao et al. (2024); Yang et al. (2023); Meng et al. (2025).

Unlike in the finite-dimensional setting, this generalized, functional in-context learning yet remains to be theoretically characterized. We, therefore, herein extend this line of theoretical inquiry to

---

[*]Equal contributions

characterize the continuum transformers that have been leveraged for in-context operator learning Calvello et al. (2024). Our contributions, therefore, are twofold: the insights afforded by such theoretical analysis and the development of a mathematical framework for doing such analysis, as we highlight in greater detail in the main text. In particular, our contributions are as follows:

- Proving continuum transformers perform in-context operator learning by performing gradient descent in a Reproducing Kernel Hilbert Space of operators and that the resulting in-context predictor recovers the Bayes Optimal Predictor under well-specified parameter choices of the transformer. Such proofs required the modeling of a generalized continuum attention mechanism and subsequent novel usages of a generalized representer theorem for Hilbert spaces and Gaussian measures over Hilbert spaces.

- Proving that such parameters under which continuum transformers implement operator gradient descent are minimizers of the training process of such transformers, leveraging a novel gradient flow analysis over the space of functionals on a Hilbert space.

- Empirically validating that continuum transformers perform in-context operator gradient descent upon inference with the exhibited parameters and that such parameters are recovered with transformer training across a diverse selection of operator RKHSs.

## 2 BACKGROUND

### 2.1 NEURAL OPERATORS

Neural operator methods, while more broadly applicable, most often seek to amortize the solution of a spatiotemporal PDE. Such PDEs are formally described by spatial fields that evolve over time, namely some $u : \Omega \times [0, \infty) \to \mathbb{R}$, where $x \in \Omega \subset \mathbb{R}^d$ are spatial coordinates and $t \in [0, \infty)$ is a time coordinate. While the more abstract operator learning framework can be formulated as seeking to learn a map $\widehat{\mathcal{G}} : \mathcal{A} \to \mathcal{U}$ between two function spaces $\mathcal{A}$ and $\mathcal{U}$, we are most often interested in learning time-rollout maps, in which the input and output function spaces are identical. In such cases, it is assumed a dataset of the form $\mathcal{D} := \{(u_i^{(0)}, u_i^{(T)})\}$ is available, where $u^{(0)} : \Omega \to \mathbb{R}$ is the initial condition, for which there exists some true operator $\mathcal{G}$ such that, for all $i$, $u_i^{(T)} = \mathcal{G}(u_i^{(0)})$. While many different learning-based approaches have been proposed to solve this operator learning problem, they all can be abstractly framed as seeking

$$\min_{\widehat{\mathcal{G}}} ||\widehat{\mathcal{G}} - \mathcal{G}||^2_{\mathcal{L}^2(\mathcal{U},\mathcal{U})} = \int_{\mathcal{U}} ||\widehat{\mathcal{G}}(u_0) - \mathcal{G}(u_0)||^2_{\mathcal{U}} \, du_0. \tag{1}$$

Within this broad family of operator learning, the most widely employed classes are the Deep Operator Networks (DeepONets) Lu et al. (2021; 2019); Wang et al. (2021); Kopaničáková & Karniadakis (2025) and Fourier Neural Operators (FNOs) Li et al. (2020b;c;a); Bonev et al. (2023). FNOs are parameterized as a sequence of layers of linear operators, given as kernel integral transforms, with standard intermediate ReLU nonlinearities. Formally, a single layer is then given by $\sigma(\mathcal{F}^{-1}(R_\ell \odot \mathcal{F}(u)))$, where $\mathcal{F}$ denotes the Fourier transform and $R_\ell$ the learnable kernel parameter. This layer encodes a translationally invariant kernel integral transform.

### 2.2 IN-CONTEXT LEARNING AND CONTINUUM TRANSFORMERS

The standard attention mechanism is parameterized by $\theta := \{W_k, W_q, W_v\}$, where $W_k \in \mathbb{R}^{d_k \times d}$, $W_q \in \mathbb{R}^{d_q \times d}$, and $W_v \in \mathbb{R}^{d_v \times d}$ for sequential data $X \in \mathbb{R}^{d \times n}$ Vaswani (2017). Formally,

$$\text{Attn}(X) := (W_v X) M H \left( (W_q X), (W_k X) \right), \tag{2}$$

where $M \in \mathbb{R}^{n \times n}$ is a masking matrix and $H : \mathbb{R}^{d_q \times n} \times \mathbb{R}^{d_k \times n} \to \mathbb{R}^{n \times n}$ is a nonlinear transform of key-query similarity measures. This resulting matrix $H(Q, K)$ is referred to as the "attention weights" matrix. Most often, $d_k = d_q$ and $H := \text{softmax}$, i.e. $[H(Q, K)]_{i,j} = \exp(Q_i^\top K_j)/\sum_\ell \exp(Q_i^\top K_\ell)$ Cheng et al. (2023). Transformers are repeated compositions of such attention blocks with residual connections and feedforward and normalization layers.

We now highlight the "continuum attention" mechanism proposed in Calvello et al. (2024) that generalizes transformers to operator learning; "continuum" here highlights that such an architecture

models data in its discretization-agnostic function form as with other neural operator methods. In particular, in place of $x_i \in \mathbb{R}^d$, $x_i \in \mathcal{X}$ for some Hilbert space $\mathcal{X}$. The natural generalization of the attention mechanism for function inputs then replaces the $W_k, W_q$, and $W_v$ matrices with linear operators. For a sequence $x \in \mathcal{X}^n$, the key, query, and value operators respectively map $\mathcal{W}_k : \mathcal{X}^n \to \mathcal{K}^n$, $\mathcal{W}_q : \mathcal{X}^n \to \mathcal{Q}^n$, and $\mathcal{W}_v : \mathcal{X}^n \to \mathcal{V}^n$, where $\mathcal{K}, \mathcal{Q}$, and $\mathcal{V}$ are Hilbert spaces. Notably, the original framing of continuum transformers Calvello et al. (2024) generalizes attention by considering an infinite index set, rather than considering infinite-dimensional tokens as we do here; we discuss how these two seemingly distinct notions are equivalent in Appendix A. In practice, such operators are implemented as kernel integral transforms, as $\mathcal{W}_q x_i = \mathcal{F}^{-1}(R_q \odot \mathcal{F} x_i)$, where $R_q$ is the Fourier parameterization of the query kernel, with $R_k$ and $R_v$ similarly defined for the key and value kernels. The continuum attention mechanism assumes $\mathcal{K} = \mathcal{Q}$ and defines

$$\text{ContAttn}(X) := (\mathcal{W}_v X) M \operatorname{softmax}((\mathcal{W}_q X), (\mathcal{W}_k X)) \tag{3}$$

where the interpretation of $M$ remains the same as that in Equation (2). In particular, while the key, query, and value operators are generalized, the resulting attention weights matrix still lies $\in \mathbb{R}^{n \times n}$. Transformer architectures, most notably in LLMs, have been observed to exhibit an unexpected behavior known as "in-context learning" (ICL), in which they perform few-shot learning without any explicit parameter updates but merely by having training examples in their context windows. We follow the conventions of Cheng et al. (2023) in formalizing the ICL phenomenon. We suppose the dataset $\mathcal{D} := \{(X_i, y_i)\}_{i=1}^n$ on which the transformer was trained has samples of the form

$$X_i = \begin{bmatrix} x_i^{(1)} & x_i^{(2)} & \ldots & x_i^{(n)} & x_i^{(n+1)} \\ y_i^{(1)} & y_i^{(2)} & \ldots & y_i^{(n)} & 0 \end{bmatrix} \qquad y_i = y_i^{(n+1)}. \tag{4}$$

We then suppose $y_i^{(t)} = f_i(x_i^{(t)})$, where $f_i \neq f_j$ for $i \neq j$. This is the critical difference in the in-context learning setting: in the standard setting, $f_i = f$ is fixed across samples and matches the $f'$ at test time, meaning the goal for the learner is to learn such an $f$. In ICL, however, the learning algorithm must be capable of "learning" an unseen $f'$ at inference time without parameter updates.

## 2.3 Operator Meta-Learning

Significant interest has emerged in meta-operator learning for spatiotemporal PDEs, in which a single network maps from initial conditions to final states across system specifications Wang et al. (2022); Zhang (2024); Sun et al. (2024); Liu et al. (2024); Cao et al. (2024); Chakraborty et al. (2022). Formally, unlike the traditional setting discussed in Section 2.1, here the dataset consists of sub-datasets $\mathcal{D} := \cup_i \mathcal{D}_i$, where $\mathcal{D}_i := \{(u_{i;0}^{(j)}, u_{i;T}^{(j)})\}_{j=1}^{n_i}$, for which the true operator can vary across sub-datasets, i.e. $\mathcal{G}_i$ satisfies $u_{i;T}^{(j)} = \mathcal{G}_i(u_{i;0}^{(j)}) \ \forall j \in [n_i]$ but $\mathcal{G}_i \neq \mathcal{G}_{i'}$ for $i \neq i'$. The goal is to then, given only a limited number of training samples $\widetilde{\mathcal{D}} := \{(\widetilde{u}_0^{(j)}, \widetilde{u}_T^{(j)})\}_{j=1}^{\widetilde{n}}$ with $\widetilde{n} \ll \sum_i n_i$ for a fixed, potentially unseen operator $\mathcal{G}'$, learn an approximation $\widehat{\mathcal{G}} \approx \widetilde{\mathcal{G}}$. Most often, this is done by pre-training a meta-learner $\mathcal{G}_{\text{ML}}$ on $\mathcal{D}$ and then fine-tuning $\mathcal{G}_{\text{ML}}$ on $\widetilde{\mathcal{D}}$ to arrive at $\widehat{\mathcal{G}}$.

Building off of the observed ICL of transformers, the fine-tuning of these meta-learners too can be performed via explicit weight modification or via in-context learning. An entire offshoot of meta-learning known as in-context operator networks (ICONs) has spawned from the latter Cao et al. (2024); Yang et al. (2023); Meng et al. (2025); Yang & Osher (2024). Recent works in this vein have demonstrated notable empirical performance leveraging bespoke transformer architectures built atop the continuum attention from Equation (3) Cao et al. (2024); Alkin et al. (2024); Cao et al. (2025); Cao (2021). Such behavior has been observed but has yet to be theoretically characterized.

## 3 Theoretical Characterization

### 3.1 In-Context Learning Model

For quick reference, we provide a compilation of the notation used throughout this and the remaining sections in Appendix L. We now consider a generalization of the continuum attention, paralleling that of Equation (2), allowing for more general key-query similarity measures. In particular, instead of restricting $\mathcal{Q} = \mathcal{K}$ as required for Equation (3) to be well-defined, we allow $H : \mathcal{Q}^{n+1} \times \mathcal{K}^{n+1} \to$

$(\mathcal{L}(\mathcal{V}))^{(n+1)\times(n+1)}$, where $\mathcal{L}(\mathcal{V})$ denotes the set of bounded linear operators from $\mathcal{V}$ to $\mathcal{V}$; note that the "$n+1$" convention is adopted as data is of the form given by Equation (4). This generalization subsumes the softmax form assumed in Equation (3), where for $c \in \mathbb{R}$, $c \in \mathcal{L}(V)$ is understood to be defined as $cf_v$ for $f_v \in \mathcal{V}$, from which we can similarly view $\mathrm{softmax} : \mathcal{Q}^{n+1} \times \mathcal{K}^{n+1} \to (\mathcal{L}(\mathcal{V}))^{(n+1)\times(n+1)}$. If $\mathcal{V} = \mathcal{X}$, an $m$-layer continuum transformer $\mathcal{T} : \mathcal{X}^{n+1} \to \mathcal{X}^{n+1}$ consisting of generalized continuum attention layers with residual connections is well-defined as $\mathcal{T} := \mathcal{T}_m \circ \ldots \circ \mathcal{T}_0$, where

$$X_{\ell+1} = \mathcal{T}_\ell(X_\ell) := X_\ell + \left( H(\mathcal{W}_{q,\ell} X_\ell, \mathcal{W}_{k,\ell} X_\ell) \mathcal{M}(\mathcal{W}_{v,\ell} X_\ell)^T \right)^T, \tag{5}$$

where we now view $\mathcal{M} : \mathcal{V}^{n+1} \to \mathcal{V}^{n+1}$ as a mask operator acting on the value functions. Note that we present this using a "double-transpose" notation as compared to Equation (2) to follow the mathematical convention of writing an operator to the left of the function it is acting upon. Notably, in the setting of in-context learning for PDEs, the pairs $(f^{(j)}, u^{(j)})$ are generally time rollout pairs $(f^{(i)}, u^{(i)}) = (u^{((i-1)\Delta t)}, u^{(i\Delta t)})$ for some time increment $\Delta t$, elaborated upon more in Appendix B. We, thus, assume $f^{(i)}, u^{(i)} \in \mathcal{X}$, that is, that they lie in a single Hilbert space. Following the formalization established in Cole et al. (2024), we then seek to learn in-context a map $\mathcal{X} \to \mathcal{X}$, for which we construct a context window consisting of training pairs $z^{(i)} \in \mathcal{Z} = \mathcal{X} \oplus \mathcal{X}$:

$$Z_0 = \begin{pmatrix} f^{(1)} & f^{(2)} & \cdots & f^{(n)} & f^{(n+1)} \\ u^{(1)} & u^{(2)} & \cdots & u^{(n)} & 0 \end{pmatrix} \in \begin{pmatrix} \mathcal{X}^{n+1} \\ \mathcal{X}^{n+1} \end{pmatrix}, \tag{6}$$

where $\{(f^{(i)}, u^{(i)})\}_{i=1}^n$ are $n$ input-output pairs, $z^{(i)} := (f^{(i)}, u^{(i)})$, and $\mathcal{T} : \mathcal{Z}^{n+1} \to \mathcal{Z}^{n+1}$. We predict $u^{(n+1)}$ as $-[\mathcal{T}(z)]_{2,n+1}$. As in the typical ICL formalization, the key, query, and nonlinearity operators only act upon the input functions. That is, denoting $X_0 = [f^{(1)}, f^{(2)}, \ldots, f^{(n)}, f^{(n+1)}] \in \mathcal{X}^{n+1}$ as the vector of input functions and $X_\ell \in \mathcal{X}^{n+1}$ similarly as the first row of the matrix $Z_\ell$,

$$Z_{\ell+1} = Z_\ell + \left( \widetilde{H}(\mathcal{W}_{q,\ell} X_\ell, \mathcal{W}_{k,\ell} X_\ell) \mathcal{M}(\mathcal{W}_{v,\ell} Z_\ell)^T \right)^T, \tag{7}$$

with the layer $\ell$ query and key block operators $\mathcal{W}_{q,\ell} : \mathcal{X}^{n+1} \to \mathcal{Q}^{n+1}$ and $\mathcal{W}_{k,\ell} : \mathcal{X}^{n+1} \to \mathcal{K}^{n+1}$ only acting on the restricted input space and $\mathcal{W}_{v,\ell} : \mathcal{Z}^{n+1} \to \mathcal{Z}^{n+1}$ on the full space. The mask block operator matrix $\mathcal{M}$ is set as $\begin{bmatrix} I_{n\times n} & 0 \\ 0 & 0 \end{bmatrix}$, with $I_{n\times n}$ denoting the $n \times n$ block identity operator. We denote the *negated* in-context prediction for an input $f^{(n+1)} := f$ after $\ell$ layers by

$$\mathcal{T}_\ell(f; (\mathcal{W}_v, \mathcal{W}_q, \mathcal{W}_k) | z^{(1)}, \ldots, z^{(n)}) := [Z_\ell]_{2,n+1} \tag{8}$$

The in-context loss is given as follows, where we critically note that the plus sign arises from the convention that the final ICL prediction carries a negative:

$$L(\mathcal{W}_v, \mathcal{W}_q, \mathcal{W}_k) = \mathbb{E}_{Z_0, u^{(n+1)}} \left( \| [Z_{m+1}]_{2,n+1} + u^{(n+1)} \|_\mathcal{X}^2 \right) \tag{9}$$

Notably, the very modeling of the generalized continuum layer in Equation (5) by introducing $H : \mathcal{Q}^{n+1} \times \mathcal{K}^{n+1} \to (\mathcal{L}(\mathcal{V}))^{(n+1)\times(n+1)}$ was a non-obvious generalization of Equation (3) in introducing this *operator-valued* nonlinearity. The more natural thing would have been to simply consider a *scalar-valued* function "similarity" measure, i.e. defining $H : \mathcal{Q}^{n+1} \times \mathcal{K}^{n+1} \to \mathbb{R}^{(n+1)\times(n+1)}$. However, critically, doing so does *not* lend itself to characterization in an RKHS, upon which our analysis over the remaining section is based. Thus, the mathematical framing of this problem as above is a worthwhile contribution that the broader community can use for further analysis.

## 3.2 IN-CONTEXT LEARNING OCCURS VIA GRADIENT DESCENT

We now show that continuum transformers *can* perform in-context learning by implementing operator gradient descent. For in-context samples $\{(f^{(i)}, u^{(i)})\}_{i=1}^n$, the in-context task is to find $O^* := \min_{O \in \mathcal{O}} L(O) := \min_{O \in \mathcal{O}} \sum_{i=1}^n \| u^{(i)} - O f^{(i)} \|_\mathcal{X}^2$. Naturally, such an operator learning problem can be solved by iteratively updating an estimate $O_{\ell+1} = O_\ell - \eta_\ell \nabla L(O_\ell)$. Our result, therefore, roughly states that the prediction of $u^{(n+1)}$ by the $\ell$-th layer of a continuum transformer, with particular choices of key, query, and value parameters, is exactly $O_\ell f$ for any test function

$f$. We defer the full proof of the theorem to Appendix E but here highlight novelties of the proof strategy in demonstrating this result. The strategy involves deriving an explicit form of the operator gradient descent steps and then showing, under a specific choice of $(\mathcal{W}_v, \mathcal{W}_q, \mathcal{W}_k)$, inference through a layer of the continuum transformer recovers this explicitly computed operator gradient expression (see Appendix D). While the general strategy parallels that of Cheng et al. (2023), the explicit construction of the gradient descent expression over operator spaces is not as directly evident as it is over finite-dimensional vector spaces. In particular, we had to invoke a generalized form of the Representer Theorem to obtain this result, as compared to its classical form leveraged therein.

**Theorem 3.1.** *Let $\kappa : \mathcal{X} \times \mathcal{X} \to \mathcal{L}(\mathcal{X})$ be an arbitrary operator-valued kernel and $\mathcal{O}$ be the operator RKHS induced by $\kappa$. Let $\{(f^{(i)}, u^{(i)})\}_{i=1}^n$ and $L(O) := \sum_{i=1}^n \|u^{(i)} - Of^{(i)}\|_{\mathcal{X}}^2$. Let $O_0 = \mathbf{0}$ and let $O_\ell$ denote the operator obtained from the $\ell$-th operator-valued gradient descent sequence of $L$ with respect to $\| \cdot \|_{\mathcal{O}}$ as defined in Equation (23). Then there exist scalar step sizes $r'_0, \ldots, r'_m$ such that if, for an $m$-layer continuum transformer $\mathcal{T} := \mathcal{T}_m \circ \ldots \circ \mathcal{T}_0$, $[\widetilde{H}(U, W)]_{i,j} = \kappa(u^{(i)}, w^{(j)})$,
$\mathcal{W}_{v,\ell} = \begin{pmatrix} 0 & 0 \\ 0 & -r'_\ell I \end{pmatrix}$, $\mathcal{W}_{q,\ell} = I$, and $\mathcal{W}_{k,\ell} = I$ for each $\ell = 0, \ldots, m$, then for any $f \in \mathcal{X}$,*

$$\mathcal{T}_\ell(f; (\mathcal{W}_v, \mathcal{W}_q, \mathcal{W}_k)|z^{(1)}, \ldots, z^{(n)}) = -O_\ell f.$$

### 3.3 In-Context Learning Recovers the Best Linear Unbiased Predictor

In the finite-dimensional setting, it was shown in Cheng et al. (2023) that if $\{y^{(j)}\}$ are outputs from the marginal of a Gaussian process with kernel $\kappa(\cdot, \cdot)$ for inputs $\{x^{(j)}\}$, a transformer of depth $m \to \infty$ with $[H(U, W)]_{i,j} = \kappa(u^{(i)}, w^{(j)})$ recovers the Bayes optimal predictor in-context for a window of $\{(x^{(j)}, y^{(j)})\}$. Roughly speaking, this result demonstrates the optimality of recovering the true $f$ in-context if the distribution on the function space from which $f$ was sampled is Gaussian. We, therefore, seek to establish a similar result for the setting of operator-valued kernels, which requires defining a Gaussian measure on the operator space from which the true $O$ is sampled.

Notably, paralleling how the finite marginals of finite-dimensional Gaussian processes induce distributions over functions, characterizing the finite marginals of a Hilbert space-valued Gaussian process induces a distribution over operators. Doing so, however, requires the formalization of a Gaussian measure on Hilbert spaces. A Hilbert space-valued random variable $F$ is said to be Gaussian if $\langle \varphi, F \rangle_{\mathcal{X}}$ is Gaussian for any $\varphi \in \mathcal{X}$ Menafoglio & Petris (2016). We now present the generalization of Gaussian processes to Hilbert spaces from Jorgensen & Tian (2024).

**Definition 3.2.** $\kappa$ **Gaussian Random Variable in Hilbert Space** (Definition 4.1 of Jorgensen & Tian (2024)) Given an operator-valued kernel $\kappa : \mathcal{X} \times \mathcal{X} \to \mathcal{L}(\mathcal{X})$, we say $U|F \sim \mathcal{N}(\mathbf{0}, \mathbf{K}(F))$, where $U = [u^{(1)}, \ldots, u^{(n+1)}]$, $F = [f^{(1)}, \ldots, f^{(n+1)}]$ and $[\mathbf{K}(F)]_{i,j} := \kappa(f^{(i)}, f^{(j)})$ if, for all $v^{(1)}, v^{(2)} \in \mathcal{X}$ and $(f^{(i)}, u^{(i)}), (f^{(j)}, u^{(j)}) \in \mathcal{X}^2$,

$$\mathbb{E}[\langle v^{(1)}, u^{(i)} \rangle_{\mathcal{X}} \langle v^{(2)}, u^{(j)} \rangle_{\mathcal{X}}] = \langle v^{(1)}, \kappa(f^{(i)}, f^{(j)}) v^{(2)} \rangle_{\mathcal{X}} \tag{10}$$

$$\langle v^{(1)}, u^{(i)} \rangle_{\mathcal{X}} \sim \mathcal{N}\left(0, \langle v^{(1)}, \kappa(f^{(i)}, f^{(i)}) v^{(1)} \rangle_{\mathcal{X}}\right). \tag{11}$$

Such Hilbert-space valued GPs have been leveraged in Hilbert-space kriging, notably studied in Menafoglio & Petris (2016); Luschgy (1996). They proved that the Best Linear Unbiased Predictor (BLUP) precisely coincides with the Bayes optimal predictor in the MSE sense, meaning the $f : \mathcal{X}^n \to \mathcal{X}$ amongst all measurable functions that minimizes $\mathbb{E}[\|X_{n+1} - f(\mathbf{X})\|_{\mathcal{X}}^2]$, for zero-mean jointly Gaussian random variables $\mathbf{X} = (X_1, \ldots, X_n) \in \mathcal{X}^n$ and $X_{n+1} \in \mathcal{X}$, is the BLUP. Their formal statement is provided for reference in Appendix F.1.

We now show that the in-context learned operator is the Best Linear Unbiased Predictor as $m \to \infty$ and thus, by Theorem F.1, Bayes optimal if the observed data are $\kappa$ Gaussian random variables and if $\widetilde{H}$ matches $\kappa$. We again highlight here the approach and main innovations in establishing this result and defer the full presentation to Appendix F.2. Roughly, the approach relied on establishing that the infinite composition $\mathcal{T}_\infty := \ldots \circ \mathcal{T}_m \circ \ldots \circ \mathcal{T}_0$ converges to a well-defined operator and that this matches the BLUP known from Hilbert space kriging literature. The careful handling of this compositional convergence and connection to Hilbert space kriging are novel technical contributions.

**Proposition 3.3.** *Let $F = [f^{(1)}, \ldots, f^{(n+1)}], U = [u^{(1)}, \ldots, u^{(n+1)}]$. Let $\kappa : \mathcal{X} \times \mathcal{X} \to \mathcal{L}(\mathcal{X})$ be an operator-valued kernel. Assume that $U|F$ is a $\kappa$ Gaussian random variable per Definition 3.2. Let the activation function $\widetilde{H}$ of the attention layer be defined as $[\widetilde{H}(U, W)]_{i,j} := \kappa(u^{(i)}, w^{(j)})$. Consider the operator gradient descent in Theorem 3.1. Then as the number of layers $m \to \infty$, the continuum transformer's prediction at layer $m$ approaches the Best Linear Unbiased Predictor that minimizes the in-context loss in Equation (9).*

### 3.4 PRE-TRAINING CAN CONVERGE TO GRADIENT DESCENT PARAMETERS

We now demonstrate that the aforementioned parameters that result in operator gradient descent from Theorem 3.1 are in fact minimizers of the continuum transformer training objective. This, in turn, suggests that under such training, the continuum transformer parameters will converge to those exhibited in the previous section. Similar results have been proven for the standard Transformer architecture, such as in Cheng et al. (2023). Demonstrating our results, however, involved highly non-trivial changes to their proof strategy. In particular, defining gradient flows over the space of functionals of a Hilbert space requires the notion of Frechet-differentiability, which is more general than taking derivatives with respect to a matrix. Additionally, rewriting the training objective with an equivalent expectation expression (see Equation (39)) required careful manipulation of the covariance operators of the data distributions as described in the symmetry discussion that follows.

The formal statement of this result is given in Theorem 3.6. This result characterizes a stationary point of the optimization problem in Equation (12) when the value operators $W_{v,\ell}$ have the form $\begin{bmatrix} 0 & 0 \\ 0 & r_\ell I \end{bmatrix}$ and the $W_{q,\ell}$ and $W_{k,\ell}$ operators have a form relating to the symmetry of the data distribution, as fully described below. If the symmetry is characterized by a self-adjoint invertible operator $\Sigma$, we establish that there exists a fixed point of the form $W_{q,\ell} = b_\ell \Sigma^{-1/2}$ and $W_{k,\ell} = c_\ell \Sigma^{-1/2}$ for some constants $\{b_\ell\}$ and $\{c_\ell\}$. This fixed point relates to Section 3.2 and Section 3.3, since if $\Sigma = I$, we recover the parameter configuration under which functional gradient descent is performed.

This proof relies on some technical assumptions on the $F$ and $U|F$ distributions and transformer nonlinearity; such assumptions are typical of such optimization analyses, as seen in related works such as Cheng et al. (2023); Ahn et al. (2023); Dutta & Sra (2024). As mentioned, the proof proceeds by studying gradient flow dynamics of the $\mathcal{W}_{k,\ell}$ and $\mathcal{W}_{q,\ell}$ operators over the $\mathcal{P}(F, U)$ distribution. Direct analysis of such gradient flows, however, becomes analytically unwieldy under unstructured distributions $\mathcal{P}(F, U)$. We, therefore, perform the analysis in a frame of reference rotated by $\Sigma^{-1/2}$. By Assumption G.3, this rotated frame preserves the $\mathcal{P}(F, U)$ distribution and by Assumption G.4 also the attention weights computed by the continuum transformer, allowing us to make conclusions on the original setting after performing the analysis in this rotated coordinate frame. We demonstrate in Section 5.3 that common kernels satisfy Assumption G.4. We defer the full proof to Appendix G.

**Assumption 3.4.** (Rotational symmetry) Let $P_F$ be the distribution of $F = [f^{(1)}, \ldots, f^{(n+1)}]$ and $\mathbb{K}(F) = \mathbb{E}_{U|F}[U \otimes U]$. We assume that there exists a self-adjoint, invertible operator $\Sigma : \mathcal{X} \to \mathcal{X}$ such that for any unitary operator $\mathcal{M}$, $\Sigma^{1/2} \mathcal{M} \Sigma^{-1/2} F \overset{d}{=} F$ and $\mathbb{K}(F) = \mathbb{K}(\Sigma^{1/2} \mathcal{M} \Sigma^{-1/2} F)$.

**Assumption 3.5.** For any $F_1, F_2 \in \mathcal{X}^{n+1}$ and any operator $S : \mathcal{X} \to \mathcal{X}$ with an inverse $S^{-1}$, the function $\widetilde{H}$ satisfies $\widetilde{H}(F_1, F_2) = \widetilde{H}(S^* F_1, S^{-1} F_2)$.

**Theorem 3.6.** *Suppose Assumption 3.4 and Assumption 3.5 hold. Let $f(r, \mathcal{W}_q, \mathcal{W}_k) := L\left(\mathcal{W}_{v,\ell} = \left\{\begin{bmatrix} 0 & 0 \\ 0 & r_\ell I \end{bmatrix}\right\}_{\ell=0,\ldots,m}, \mathcal{W}_{q,\ell}, \mathcal{W}_{k,\ell}\right)$, where $L$ is as defined in Equation (9). Let $\mathcal{S} \subset \mathcal{O}^{m+1} \times \mathcal{O}^{m+1}$ denote the set of $(\mathcal{W}_q, \mathcal{W}_k)$ operators with the property that $(\mathcal{W}_q, \mathcal{W}_v) \in \mathcal{S}$ if and only if for all $\ell \in \{0, \ldots, m\}$, there exist scalars $b_\ell, c_\ell \in \mathbb{R}$ such that $\mathcal{W}_{q,\ell} = b_\ell \Sigma^{-1/2}$ and $\mathcal{W}_{k,\ell} = c_\ell \Sigma^{-1/2}$. Then*

$$\inf_{(r, (\mathcal{W}_q, \mathcal{W}_k)) \in \mathbb{R}^{m+1} \times \mathcal{S}} \sum_{\ell=0}^{m} \left[ (\partial_{r_\ell} f)^2 + \|\nabla_{\mathcal{W}_{q,\ell}} f\|_{\mathrm{HS}}^2 + \|\nabla_{\mathcal{W}_{k,\ell}} f\|_{\mathrm{HS}}^2 \right] = 0. \tag{12}$$

*Here $\nabla_{\mathcal{W}_{q,\ell}}$ and $\nabla_{\mathcal{W}_{k,\ell}}$ denote derivatives with respect to the Hilbert-Schmidt norm $\| \cdot \|_{\mathrm{HS}}$.*

## 4 RELATED WORKS

We were interested herein in providing a theoretical characterization of the in-context learning exhibited by continuum attention-based ICONs, paralleling that done for finite-dimensional transformers Akyürek et al. (2022); Garg et al. (2022); Dai et al. (2022). While previous works have yet to formally characterize ICL for continuum transformers, a recent work Cole et al. (2024) began a line of inquiry in this direction. Their work, however, studied a fundamentally distinct aspect of functional ICL, characterizing the sample complexity and resulting generalization for linear elliptic PDEs.

Most relevant in the line of works characterizing finite-dimensional ICL is Cheng et al. (2023). Loosely speaking, they demonstrated that, if a kernel $\kappa(\cdot, \cdot)$ is defined with $\mathbb{H}$ denoting its associated RKHS and a transformer is then defined with a specific $W_k, W_q, W_v$ and $[H(Q, K)]_{i,j} = \kappa(Q_i, K_j)$ from Equation (2), an inference pass through $m$ layers of such a transformer predicts $y_i^{(n+1)}$ from Equation (4) equivalently to $\widehat{f}(x_i^{(n+1)})$, where $\widehat{f}$ is the model resulting from $m$ steps of functional gradient descent in $\mathbb{H}$. They then demonstrated that the predictor learned by such ICL gradient descent is Bayes optimal under particular circumstances. These results are provided in Appendix C.

Notably, these results required non-trivial changes to be generalized to operator RKHSs as we did herein; as discussed, such mathematical tooling is more general than its application to the setting considered herein. In particular, our approach to performing the analysis in the infinite-dimensional function space directly without requiring discretization required formalizing several notions to rigorously justify being able to lift the proofs from finite to infinite dimensions. We view this as a significant contribution to the community. Now that we have gone through this formalism, the broader community can directly use these analysis strategies to further study in-context learning or unrelated inquiries. This is highly valuable, since it suggests that theoreticians can often follow their finite-dimensional intuitions and defer to our infinite-dimensional results to rigorously justify their results. In other words, our proof strategies suggest to other researchers working on similar problems that they need not be bogged down in the details of the error convergence minutiae that often arise in approaches relying on finite projections and can instead work with these clean abstractions.

Similarly, this work opens up the space for who can contribute to further the theoretical study of operator ICL. Much of the classical optimization community, for instance, may not be intimately familiar with the mathematical formalisms required around operator spaces. However, with our formalism, they can provide insights with little change from how they would approach finite-dimensional analyses. We, therefore, believe this framework, consisting of the generalized transformer and characterization of its optimization, is a worthwhile contribution to the theory community.

## 5 EXPERIMENTS

We now wish to empirically verify the theoretically demonstrated claims. We provide setup details in Section 5.1 and then study the claims of Section 3.3 in Section 5.2 and those of Section 3.4 in Section 5.3. Code for this verification is available at https://github.com/yashpatel5400/opicl.

### 5.1 EXPERIMENTAL SETUP

In the experiments that follow, we wish to draw $\kappa$ Gaussian Random Variables, as per Definition 3.2. To begin, we first define an operator-valued covariance kernel $\kappa$. In particular, we consider one commonly encountered in the Bayesian functional data analysis literature Kadri et al. (2016a), namely the Hilbert-Schmidt Integral Operator. Suppose $k_x : \mathcal{X} \times \mathcal{X} \to \mathbb{R}$ and $k_y : \Omega \times \Omega \to \mathbb{R}$ are both positive definite kernels, where $k_y$ is Hermitian. Then, the following is a well-defined kernel:

$$[\kappa(f^{(1)}, f^{(2)})u](y) := k_x(f^{(1)}, f^{(2)}) \int k_y(y', y)u(y')dy'. \tag{13}$$

Notably, similar to how functions in a scalar RKHS can be sampled as $f = \sum_{s=1}^{S} \alpha^{(s)} k(x^{(s)}, \cdot)$ for $\alpha^{(s)} \sim \mathcal{N}(0, \sigma^2)$ over a randomly sampled collection $\{x^{(s)}\}_{s=1}^{S}$, we can sample operators by sampling a collection $\{(\varphi^{(s)}, \psi^{(s)})\}_{s=1}^{S}$ and defining $O = \sum_{i=s}^{S} \alpha^{(s)} \left(k_x(\varphi^{(s)}, \cdot) \int k_y(y', y)\psi^{(s)}(y')dy'\right)$ with $\alpha^{(s)} \sim \mathcal{N}(0, \sigma^2)$. We focus on $\mathcal{X} = \mathcal{L}^2(\mathbb{T}^2)$,

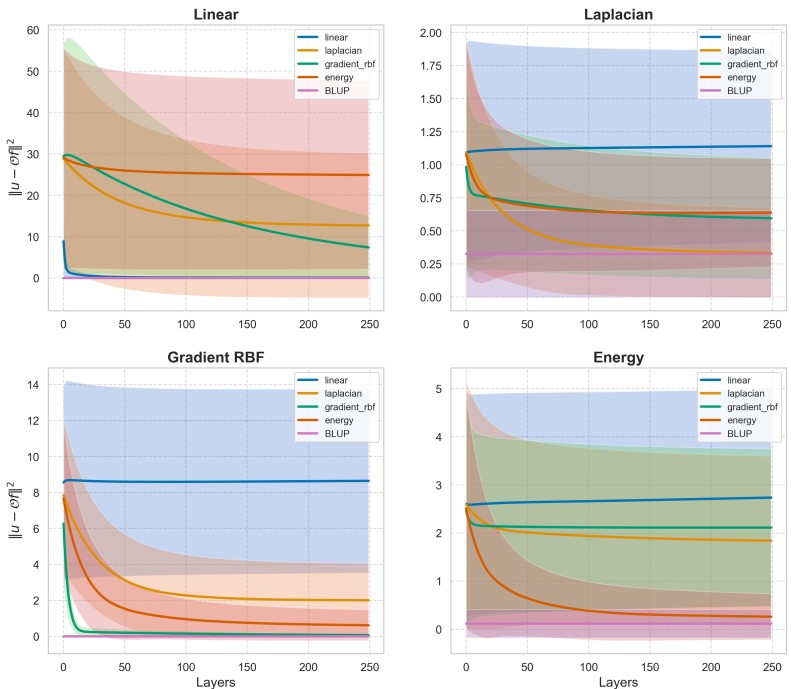

Figure 1: In-context learning loss curves over the number of layers in the continuum transformer. The kernels of the data-generating processes are given in the titles of the sub-figures. Curves show the mean $\pm 1/2$ standard deviation from 50 i.i.d. draws of the operator.

for which the functions $\varphi^{(s)}, \psi^{(s)}$ can be sampled from a Gaussian with mean function 0 and covariance operator $\alpha(-\Delta + \beta I)^{-\gamma}$, where $\alpha, \beta, \gamma \in \mathbb{R}$ are parameters that control the smoothness of the sampled functions and $\Delta$ is the Laplacian operator. Such a distribution is typical of the neural operators literature, as seen in Subedi & Tewari (2025); Kovachki et al. (2021); Bhattacharya et al. (2021), and is sampled as

$$\varphi^{(s)} := \sum_{|\nu|_\infty \leq N/2} \left( Z_\nu^{(s)} \alpha^{1/2} (4\pi^2 ||\nu||_2^2 + \beta)^{-\gamma/2} \right) e^{i\nu \cdot x} \quad \text{where } Z_\nu^{(s)} \sim \mathcal{N}(0,1), \quad (14)$$

where $N/2$ is the Nyquist frequency assuming a discretized spatial resolution of $N \times N$. Such sampling is similarly repeated for $\psi^{(s)}$. We consider standard scalar kernels $k_x$ and $k_y$ to define Hilbert-Schmidt kernels. For $k_x$, we consider the Linear, Laplacian, Gradient RBF, and Energy kernels and for $k_y$, the Laplace and Gaussian kernels. The definitions of such kernels is deferred to Appendix H. To finally construct the in-context windows, we similarly sample functions $f^{(j)}$ per Equation (14) and evaluate $u^{(j)} = Of^{(j)}$ using the sampled operator. To summarize, a *single* ICL context window $i$ of the form Equation (6) is constructed by defining an operator $O_i$ with a random sample $\{(\alpha_i^{(s)}, \varphi_i^{(s)}, \psi_i^{(s)})\}_{s=1}^S$, sampling input functions $\{f_i^{(j)}\}$, and evaluating $u_i^{(j)} = O_i f_i^{(j)}$.

## 5.2 BEST LINEAR UNBIASED PREDICTOR EXPERIMENTS

We now empirically demonstrate the claim of Proposition 3.3, specifically using the Hilbert-Schmidt operator-valued kernels described in the previous section. In particular, we demonstrate the optimality of the continuum transformer in-context update steps if the nonlinearity is made to match the kernel of the data-generating process. We consider four pairs of the aforementioned $(k_x, k_y)$ to define the operator-valued kernels and then fix the model parameters to be those given by Theorem 3.1. Notably, as $\mathcal{W}_{k,\ell}$ and $\mathcal{W}_{q,\ell}$ are implemented as FNO kernel integral layers, we do so by fixing $R_{k,\ell} = R_{q,\ell} = \mathbb{1}_{(N/2) \times (N/2)} \ \forall \ell$. The results are shown in Figure 1. As expected, we find the in-context loss curves to decrease over increasing layers when the kernel matches the nonlinearity, as each additional layer then corresponds to an additional step of operator gradient descent per

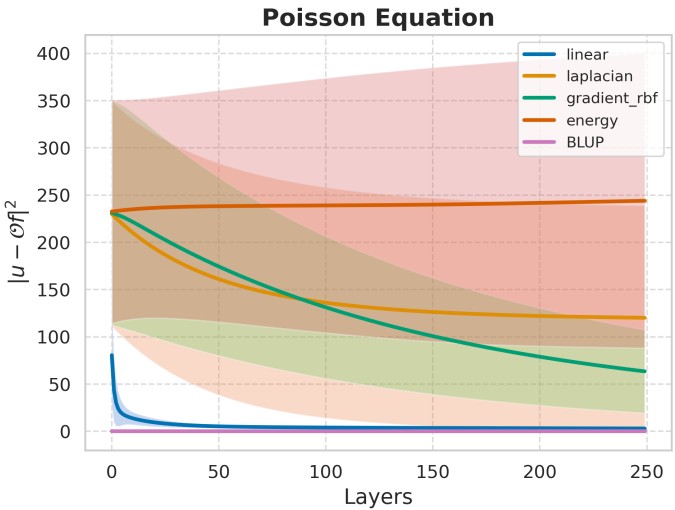

Figure 2: In-context learning loss curves over the number of layers in the continuum transformer for the Poisson equation samples. Curves show the mean $\pm 1/2$ standard deviation from 50 i.i.d. draws of the operator.

Theorem 3.1. For each setup, we also construct the BLUP to demonstrate the desired convergence, whose explicit prediction is given by

$$\mathcal{O}^*_{\text{BLUP}} f = \sum_{i,j=1}^{n} \kappa(f, f^{(i)})[\hat{\mathbb{K}}^{-1}]_{ij} u^{(j)}. \tag{15}$$

We see the optimality of matching the kernel and nonlinearity across the different choices of kernels in the limit of $\ell \to \infty$, namely in converging to the same error as the BLUP. We see that this result is robust over samplings of the operator, as the results in Figure 1 are reported over 50 independent trials. We visualize the in-context learned predictions for each setup combination in Appendix I, which reveals that, when $k_x$ matches the data-generating kernel, the resulting field predictions structurally match the true $\mathcal{O} f$.

### 5.2.1 POISSON EQUATION

In the previous experiment, we demonstrated the optimality of the estimator when its kernel matches that of the data-generating RKHS. In practical settings, however, selecting $\kappa$ in this fashion may not be possible, as estimating the kernel of the RKHS from which samples are drawn is often difficult. We, therefore, now study the robustness of this exhibited behavior in a realistic PDE learning setting.

In particular, we study the 2D Poisson equation. The 2D Poisson equation is given by $\Delta u(x) = f(x)$, where $u : \mathbb{T}^2 \to \mathbb{R}$ is the scalar field of interest and $f : \mathbb{T}^2 \to \mathbb{R}$ is the source field. In this setting, we wish to learn the solution operator $\mathcal{G} : f \to u$. The $f$ source fields were again drawn from a GRF, as described in the Section 5.1, and corresponding solutions $u$ computed using an analytic FFT-based Poisson solver. Representative samples are visualized in Appendix J.

The results are shown in Figure 2. We see that the exhibited parameters of the continuum transformer continue to display the desired optimization characteristics over layers even in this case where the explicit kernel is unknown. In particular, we find the linear kernel to exhibit optimal performance here; future work exploring the systematic selection of the optimal $\kappa$ in practical settings would be of great interest. Note that the BLUP achieves near-perfect estimation accuracy here, since the solution operator of the Poisson Equation over periodic boundary conditions is linear in $f$.

### 5.3 OPTIMIZATION EXPERIMENTS

We now seek to empirically validate Theorem 3.6, namely that $\mathcal{W}_{k,\ell} = b_\ell \Sigma^{-1/2}$ and $\mathcal{W}_{q,\ell} = c_\ell \Sigma^{-1/2}$ are fixed points of the optimization landscape. Direct verification of this statement, how-

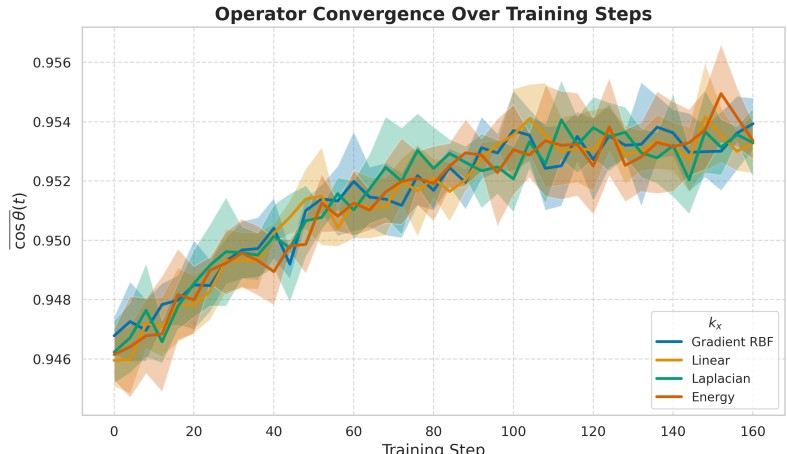

Figure 3: Pairwise convergence of the key-key, key-query, and query-query operators of the continuum transformer in Hilbert-Schmidt cosine similarity across different kernels $k_x$ over training steps. Curves show the mean $\pm 1$ standard deviation from 5 i.i.d. trials of training procedure.

ever, is not feasible, since do not have an explicit form of the $\Sigma^{-1/2}$ operator. Nonetheless, we can verify that, for any $\ell_1, \ell_2$ and $i, j \in \{q, k\}$ (indicating whether we are comparing a key-key, query-query, or key-query operator pair), $\langle \overline{\mathcal{W}}_{i,\ell_1}, \overline{\mathcal{W}}_{j,\ell_2} \rangle_{\mathrm{HS}} \to 1$, where $\overline{\mathcal{W}} := \mathcal{W}/\|\mathcal{W}\|_{\mathrm{HS}}$ denotes the normalized operator. Since we are working over the space $\mathcal{X} = \mathcal{L}^2(\mathbb{T}^2)$ and considering Fourier kernel parameterization for the operators, this Hilbert-Schmidt inner product can be computed over the kernels, i.e. $\langle \overline{R}_{i,\ell_1}, \overline{R}_{j,\ell_2} \rangle_{\mathrm{F}}$, where $\langle \cdot, \cdot \rangle_{\mathrm{F}}$ is the Frobenius inner product. The final metric is then

$$\overline{\cos\theta}(t) := \frac{1}{4m^2} \sum_{i,j \in \{k,q\}} \sum_{\ell_1, \ell_2 \in \{1,\dots,m\}} \frac{\langle R_{i,\ell_1}, R_{j,\ell_2} \rangle_{\mathrm{F}}}{\|R_{i,\ell_1}^{(t)}\|_F \|R_{j,\ell_2}^{(t)}\|_F}, \tag{16}$$

representing the *average* pairwise cosine similarity at step $t$ of the optimization between the learned operators. Since we again consider a 250-layer continuum transformer, naively computing Equation (16) is computationally expensive; we, therefore, report this metric over a random sampling of $m' = 10$ layers of the network. We repeat this optimization procedure over 5 independent trials, where randomization occurs in the sampling of the datasets across trials and network initializations.

As required by Theorem 3.6, we constrain the optimization of $\mathcal{W}_{v,\ell}$ to be over operators of the form $\begin{bmatrix} 0 & 0 \\ 0 & r_\ell \end{bmatrix}$. This procedure is repeated across each of the $k_x$ kernels considered in the previous section with $k_y$ fixed to be the Gaussian kernel; we demonstrate that the Linear $k_x$ kernel satisfies Assumption G.4 in Appendix K.1. Notably, the other choices of $k_x$ do not satisfy this assumption, yet we find that the convergence result holds robustly to this violation. Specific hyperparameter choices of the training are presented in Appendix K. The results are presented in Figure 3. As mentioned, we see that the operators all converge pairwise on average, validating the characterization of the fixed points given by Theorem 3.6. As in the previous experiment, we find results to consistently replicate across training runs, pointing to this finding being robust to initializations.

## 6 DISCUSSION

In this paper, we provided a theoretical characterization of the ICL phenomenon exhibited by continuum transformers and further validated such claims empirically. Unlike in the language learning context, this insight suggests a practical direction for improving ICL for meta-learning in PDEs, namely by estimating $\kappa$ for specific PDE meta-learning tasks and using this directly to parameterize $\widetilde{H}$ in the continuum transformer architecture. Such RKHSs are often not explicitly defined but rather induced by distributions on parameters of the PDE and its parametric form. Additionally, the results of Section 5.3 suggest that a stronger version of Theorem 3.6 should hold, in which the convergence result is independent of $\ell$; proving such a generalization would be of great interest.

## 7 ACKNOWLEDGMENT

The authors thank Julian Bernado for his input in the early stages of this work. Ambuj Tewari acknowledges the support of NSF via grant DMS-2413089.

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

## A    CONTINUUM TRANSFORMER TOKEN DETAILS

In Calvello et al. (2024), the setting of interest was in extending the standard vision transformers that typically act on finite-dimensional images $\mathbb{R}^{H \times W}$ to instead act on a continuum, i.e. over a function mapping $\Omega \to \mathbb{R}$ for $\Omega \subset \mathbb{R}^2$. In doing so, they had to generalize the typical notions of patching that are introduced in vision transformers, by considering patches that decompose the domain, i.e. $\Omega_i$ such that $\cup_i \Omega_i = \Omega$ and $\Omega_i \cap \Omega_j = \emptyset$. From here, each patch becomes a separate function $f_i : \Omega_i \to \mathbb{R}$, which necessitated dealing with infinite-dimensional tokens in their architecture, as we assumed directly for our setting.

## B    TIME ROLLOUT META LEARNING

As discussed in the main text, the pairs $(f^{(j)}, u^{(j)})$ are generally time rollout pairs $(f^{(j)}, u^{(j)}) = (u^{((j-1)\Delta t)}, u^{(j\Delta t)})$. We elaborate on this common setting below. In the context of doing time-rollouts, a pre-training dataset of the form $\mathcal{D} := \{(U_i^{(0:T-1)}, U_i^{(T)})\}$ is available, where $U_i^{(0:T)} := [u_i^{(0)}, u_i^{(\Delta T)}, ..., u_i^{((T-1)\Delta T)}]$, with $u_i^{(t)} \in \mathcal{L}^2(\Omega)$. Notably, such a setup is equivalent to having $n = T/\Delta T$ training pairs $\{(u_i^{((t-1)\Delta T)}, u_i^{t\Delta T})\}_{t=1}^n$. It is further assumed that, for each sample $i$, there is a true, deterministic solution operator $\mathcal{G}_i \in \mathcal{O}$, where $\mathcal{O}$ is a space of operators, that maps from the spatial field at some time $t$ to its state at some later $t + \Delta T$. The in-context learning goal, thus, is, given a new sequence of $\widetilde{U}^{(0:T-1)}$ generated by some unseen operator $\widetilde{\mathcal{G}}$, predict $\widetilde{U}^{(T)}$, i.e.

$$Z_0 = \begin{pmatrix} \tilde{u}^{(0)} & \tilde{u}^{(\Delta t)} & \dots & \tilde{u}^{((n-1)\Delta t)} & \tilde{u}^{(n\Delta t)} \\ \tilde{u}^{(\Delta t)} & \tilde{u}^{(2\Delta t)} & \dots & \tilde{u}^{(n\Delta t)} & 0 \end{pmatrix}$$

## C    RKHS FUNCTIONAL GRADIENT DESCENT THEOREMS

We provide here the precise statements of the relevant results from Cheng et al. (2023).

**Proposition C.1.** *(Proposition 1 from Cheng et al. (2023)) Let $\mathcal{K}$ be an arbitrary kernel. Let $\mathcal{H}$ denote the Reproducing Kernel Hilbert space induced by $\mathcal{K}$. Let $\mathbf{z}^{(i)} = (x^{(i)}, y^{(i)})$ for $i = 1, \dots, n$ be an arbitrary set of in-context examples. Denote the empirical loss functional by*

$$L(f) := \sum_{i=1}^n \left( f(x^{(i)}) - y^{(i)} \right)^2. \tag{17}$$

*Let $f_0 = 0$ and let $f_\ell$ denote the gradient descent sequence of $L$ with respect to $\| \cdot \|_{\mathcal{H}}$, as defined in (3.1). Then there exist scalars stepsizes $r_0, \dots, r_m$ such that the following holds:*

*Let $\widetilde{H}$ be the function defined as*

$$\widetilde{H}(U, W)_{i,j} := \mathcal{K}(U^{(i)}, W^{(j)}), \tag{18}$$

*where $U^{(i)}$ and $W^{(j)}$ denote the ith column of $U$ and $W$ respectively. Let*

$$V_\ell = \begin{bmatrix} 0 & 0 \\ 0 & -r_\ell \end{bmatrix}, \quad B_\ell = I_{d \times d}, \quad C_\ell = I_{d \times d}. \tag{19}$$

*Then for any $x := x^{(n+1)}$, the Transformer's prediction for $y^{(n+1)}$ at each layer $\ell$ matches the prediction of the functional gradient sequence (3.1) at step $\ell$, i.e., for all $\ell = 0, \dots, k$,*

$$\mathcal{T}_\ell(x; (V, B, C)|z^{(1)}, \dots, z^{(n)}) = -f_\ell(x). \tag{20}$$

**Proposition C.2.** *(Proposition 2 from Cheng et al. (2023)) Let*

$$X = \begin{bmatrix} x^{(1)}, & \dots, & x^{(n+1)} \end{bmatrix}, \quad Y = \begin{bmatrix} y^{(1)}, & \dots, & y^{(n+1)} \end{bmatrix}. \tag{21}$$

*Let $\mathcal{K} : \mathbb{R}^d \times \mathbb{R}^d \to \mathbb{R}$ be a kernel. Assume that $Y|X$ is drawn from the $\mathcal{K}$ Gaussian process. Let the attention activation*

$$\widetilde{H}(U, W)_{ij} := \mathcal{K}(U^{(i)}, W^{(j)}), \tag{22}$$

*and consider the functional gradient descent construction in Proposition 1. Then, as the number of layers $L \to \infty$, the Transformer's prediction for $y^{(n+1)}$ at layer $\ell$ (2.4) approaches the Bayes (optimal) estimator that minimizes the in-context loss (2.5).*

# D  GRADIENT DESCENT IN OPERATOR SPACE

We start by some defining notation that we will use in the next sections. We denote by $\mathcal{X} = \{x : D_X \to \mathbb{R}\}$ and $\mathcal{Y} = \{y : D_Y \to \mathbb{R}\}$ the separable Hilbert spaces in which our input and output functions lie in respectively. We denote by $\mathcal{C}(\mathcal{X}, \mathcal{Y})$ the space of continuous operators from $\mathcal{X}$ to $\mathcal{Y}$. Let $\mathcal{L}(\mathcal{Y})$ denote the set of bounded linear operators from $\mathcal{Y}$ to $\mathcal{Y}$.

We begin by defining gradient descent in operator space. Let $\mathcal{O}$ denote a space of bounded operators from $\mathcal{X}$ to $\mathcal{Y}$ equipped with the operator norm $\|\cdot\|_{\mathcal{O}}$. Let $L : \mathcal{O} \to \mathbb{R}$ denote a loss function. The gradient descent of $L$ is defined as the sequence

$$O_{\ell+1} = O_\ell - r_\ell \nabla L(O_\ell) \tag{23}$$

where

$$\nabla L(O) = \underset{G \in \mathcal{O}, \|G\|_{\mathcal{O}}=1}{\arg\min} \left. \frac{d}{dt} L(O + tG) \right|_{t=0}.$$

Suppose we have $n$ input-output function pairs as $f^{(1)}, \ldots, f^{(n)} \in \mathcal{X}$ and $u^{(1)}, \ldots, u^{(n)} \in \mathcal{Y}$ and we define $L$ as the weighted empirical least-squares loss

$$L(O) = \sum_{i=1}^{n} \|u^{(i)} - Of^{(i)}\|_{\mathcal{Y}}^2.$$

Then $\nabla L(O)$ takes the form

$$\nabla L(O) = \underset{G \in \mathcal{O}, \|G\|_{\mathcal{O}}=1}{\arg\min} \left. \frac{d}{dt} \sum_{i=1}^{n} \|u^{(i)} - (O + tG)f^{(i)}\|_{\mathcal{Y}}^2 \right|_{t=0}. \tag{24}$$

For simplification, suppose that we denote by $G^*$ the steepest descent direction. Then the method of Lagrange multipliers states that there exists some $\lambda$ for which the problem in Equation (24) is equivalent to

$$G^* = \underset{G \in \mathcal{O}}{\arg\min} \left. \frac{d}{dt} \sum_{i=1}^{n} \|u^{(i)} - (O + tG)f^{(i)}\|_{\mathcal{Y}}^2 \right|_{t=0} + \lambda \|G\|_{\mathcal{B}(\mathcal{X}, \mathcal{Y})}^2 \tag{25}$$

$$= \underset{G \in \mathcal{O}}{\arg\min} \sum_{i=1}^{n} 2\langle u^{(i)} - Of^{(i)}, Gf^{(i)}\rangle_{\mathcal{Y}} + \lambda \|G\|_{\mathcal{B}(\mathcal{X}, \mathcal{Y})}^2. \tag{26}$$

The second line can be calculated by thinking of the loss function as a composition of functions $L = L_2 \circ L_1$, $L_1 : \mathbb{R} \to \mathcal{Y}$ which takes

$$L_1(t) = u^{(i)} - (O + tG)f^{(i)}$$

and $L_2 : \mathcal{Y} \to \mathbb{R}$ where

$$L_2(y) = \langle y, y \rangle.$$

Then $L_1'(t)(s) = Gu^{(i)}$ and $L_2'(y)(h) = 2\langle y, h \rangle$. We have

$$\begin{aligned}
(L_2 \circ L_1(t))'(s) &= L_2'(L_1(t)) \circ L_1'(t)(s) \\
&= L_2'(u^{(i)} - (O + tG)f^{(i)})(Gf^{(i)}) \\
&= 2\langle u^{(i)} - (O + tG)f^{(i)}, Gf^{(i)}\rangle_{\mathcal{Y}}.
\end{aligned}$$

Evaluating the derivative at $t = 0$ gives the desired expression

## D.1  GRADIENT DESCENT IN OPERATOR RKHS

We now introduce the RKHS framework on our space of operators by using an operator-valued kernel. The following definitions were posed in Kadri et al. (2016b) (Section 4, Definitions 3 and 5).

**Definition D.1. Operator-valued Kernel** An operator-valued kernel is a function $\kappa : \mathcal{X} \times \mathcal{X} \to \mathcal{L}(\mathcal{Y})$ such that

(i) $\kappa$ is Hermitian, that is, for all $f_1, f_2 \in \mathcal{X}, \kappa(f_1, f_2) = \kappa(f_2, f_1)^*$ where $^*$ denotes the adjoint operator.

(ii) $\kappa$ is positive definite on $\mathcal{X}$ if it is Hermitian and for every $n \in \mathbb{N}$ and all $(f_i, u_i) \in \mathcal{X} \times \mathcal{Y} \; \forall \; i = 1, 2, \ldots, n$, the matrix with $(i, j)$-th entry $\langle \kappa(f_i, f_j) u_i, u_j \rangle$ is a positive definite matrix.

**Definition D.2. Operator RKHS** Let $\mathcal{O}$ be a Hilbert space of operators $O : \mathcal{X} \to \mathcal{Y}$, equipped with an inner product $\langle \cdot, \cdot \rangle_{\mathcal{O}}$. We call $\mathcal{O}$ an operator RKHS if there exists an operator-valued kernel $\kappa : \mathcal{X} \times \mathcal{X} \to \mathcal{L}(\mathcal{Y})$ such that

(i) The function $g \to \kappa(f, g)u$ for $\mathcal{X}$ belongs to the space $\mathcal{O}$ for all $f \in \mathcal{X}, u \in \mathcal{Y}$.

(ii) $\kappa$ satisfies the reproducing kernel property:

$$\langle O, \kappa(f, \cdot)u \rangle_{\mathcal{O}} = \langle Of, u \rangle_{\mathcal{Y}}$$

for all $O \in \mathcal{O}, f \in \mathcal{X}, u \in \mathcal{Y}$.

We now state the Representer Theorem for operator RKHS's, as stated in Theorem 11 of Stepaniants (2023). Assume that $\mathcal{O}$ can be decomposed orthogonally into $\mathcal{O} = \mathcal{O}_0 \oplus \mathcal{O}_1$ where $\mathcal{O}_0$ is a finite-dimensional Hilbert space spanned by the operators $\{E_k\}_{k=1}^r$ and $\mathcal{O}_1$ is its orthogonal complement under the inner product $\langle \cdot, \cdot \rangle_{\mathcal{O}}$. We denote the inner product $\langle \cdot, \cdot \rangle_{\mathcal{O}}$ restricted to $\mathcal{O}_0, \mathcal{O}_1$ as $\langle \cdot, \cdot \rangle_{\mathcal{O}_0}, \langle \cdot, \cdot \rangle_{\mathcal{O}_1}$ respectively.

**Theorem D.3.** *Let $\psi : \mathbb{R} \to \mathbb{R}$ be a strictly increasing real-valued function and let $\mathcal{L}(\mathcal{X} \times \mathcal{Y} \times \mathcal{Y}) \to \mathbb{R}$ be an arbitrary loss function. Then*

$$\widehat{O} = \arg\min_{O \in \mathcal{O}} \mathcal{L}\left(\{f^{(i)}, u^{(i)}, Of^{(i)}\}_{i=1}^n\right) + \psi(\|\text{proj}_{\mathcal{O}_1} O\|_{\mathcal{O}_1})$$

*has the form*

$$\widehat{O} = \sum_{k=1}^r d_k E_k + \sum_{i=1}^n \kappa(f^{(i)}, \cdot)\alpha_i$$

*for some $d \in \mathbb{R}^r$, $E_k \in \mathcal{O}_0$ for all $k \in \{1, \ldots, r\}$ and $\alpha_i \in \mathcal{Y}$ for all $i \in \{1, \ldots, n\}$.*

We use this theorem to simplify the expression for gradient descent in operator space.

**Lemma D.4.** *Given any $O \in \mathcal{O}$, let $G^*$ denote the steepest descent direction of the weighted least-squares loss with respect to $\| \cdot \|_{\mathcal{O}}$ as given in equation 25. Suppose $\mathcal{O}$ is an RKHS with kernel $\kappa$. Then*

$$G^*(\cdot) = c \sum_{i=1}^n \kappa(f^{(i)}, \cdot)(u^{(i)} - Of^{(i)}) \tag{27}$$

*for some scalar $c \in \mathbb{R}^+$.*

*Proof.* We apply theorem D.3 to equation Equation (25) with $\mathcal{O}_0$ the trivial subspace and $\psi(s) = \frac{\lambda}{2}s^2$. Then our solution has the form

$$G^*(\cdot) = \sum_{i=1}^n \kappa(f^{(i)}, \cdot)\alpha_i. \tag{28}$$

We also know that

$$\|G^*\|_{\mathcal{O}}^2 = \sum_{i,j=1}^n \langle \kappa(f^{(i)}, \cdot)\alpha_i, \kappa(f^{(j)}, \cdot)\alpha_j \rangle_{\mathcal{O}} = \sum_{i,j=1}^n \langle \kappa(f^{(i)}, f^{(j)})\alpha_i, \alpha_j \rangle_{\mathcal{Y}}$$

where the last equality follows from the RKHS property. We observe that

$$\sum_{i,j=1}^n \langle \kappa(f^{(i)}, \cdot)\alpha_i, \kappa(f^{(j)}, \cdot)\alpha_j \rangle_{\mathcal{O}} = \sum_{i,j=1}^n \langle \alpha_i, \kappa(f^{(i)}, f^{(j)})\alpha_j \rangle_{\mathcal{Y}}$$

by the same RKHS property. Note that $\kappa(f^{(i)}, f^{(j)}) \in \mathcal{L}(\mathcal{Y})$, that is, is a linear operator from $\mathcal{Y}$ to $\mathcal{Y}$. Let $U \in \mathcal{X}^n, F \in \mathcal{Y}^n$ be such that $U_i = u^{(i)}$ and $F_i = Of^{(i)}$. Then

$$\alpha^* = \arg\min_{\alpha \in \mathcal{Y}^n} \sum_{i,j=1}^n 2\langle u^{(i)} - Of^{(i)}, \kappa(f^{(i)}, f^{(j)})\alpha_j\rangle_\mathcal{Y} + \lambda\langle\alpha_i, \kappa(f^{(i)}, f^{(j)})\alpha_j\rangle_\mathcal{Y}$$

$$= \arg\min_{\alpha \in \mathcal{Y}^n} \sum_{i,j=1}^n \langle 2(u^{(i)} - Of^{(i)} + \lambda\alpha_i), \kappa(f^{(i)}, f^{(j)})\alpha_j\rangle_\mathcal{Y}$$

Taking the gradient of $\alpha$ as zero, that is, $\nabla_\alpha = 0$ gives us $\alpha \propto U - OF$ (here we are looking at $\alpha$ as an element of $\mathcal{Y}^n$). We also note that since $\|G^*\|_\mathcal{O} = 1$,

$$\sum_{i,j=1}^n \langle\alpha_i, \kappa(f^{(i)}, f^{(j)})\alpha_j\rangle_\mathcal{Y} = 1.$$

It follows that

$$\alpha^* = \frac{1}{\sum_{i,j=1}^n \langle u^{(i)} - Of^{(i)}, \kappa(f^{(i)}, f^{(j)})(u^{(j)} - Of^{(j)})\rangle_\mathcal{Y}}(U - OF).$$

Therefore

$$G^*(\cdot) = \frac{1}{\sum_{i,j=1}^n \langle u^{(i)} - Of^{(i)}, \kappa(f^{(i)}, f^{(j)})(u^{(j)} - Of^{(j)})\rangle_\mathcal{Y}} \sum_{i=1}^n \kappa(f^{(i)}, \cdot)(u^{(i)} - Of^{(i)}).$$

This gives us an exact form of $c$ as stated in equation Equation (27). □

# E   IN-CONTEXT LEARNING VIA GRADIENT DESCENT PROOF

We first recall some notation from section 3.2. We are given $n$ demonstrations $(f^{(i)}, u^{(i)}) \in \mathcal{X} \times \mathcal{X}$ for all $i \in [n]$. We set $\mathcal{Y} = \mathcal{X}$ for our purpose. The goal is to predict the output function for $f^{(n+1)}$. We stack these in a matrix $Z_0$ that serves as the input to our transformer:

$$Z_0 = [z^{(1)}, \ldots, z^{(n)}, z^{(n+1)}] = \begin{pmatrix} f^{(1)} & f^{(2)} & \cdots & f^{(n)} & f^{(n+1)} \\ u^{(1)} & u^{(2)} & \cdots & u^{(n)} & 0 \end{pmatrix}.$$

$Z_\ell$ denotes the output of layer $\ell$ of the transformer as given in equation 7.

**Theorem E.1.** *Let $\kappa : \mathcal{X} \times \mathcal{X} \to \mathcal{L}(\mathcal{X})$ be an arbitrary operator-valued kernel and $\mathcal{O}$ be the operator RKHS induced by $\kappa$. Let $\{(f^{(i)}, u^{(i)})\}_{i=1}^n$ and $L(O) := \sum_{i=1}^n \|u^{(i)} - Of^{(i)}\|_\mathcal{X}^2$. Let $O_0 = \mathbf{0}$ and let $O_\ell$ denote the operator obtained from the $\ell$-th operator-valued gradient descent sequence of $L$ with respect to $\|\cdot\|_\mathcal{O}$ as defined in Equation (23). Then there exist scalar step sizes $r'_0, \ldots, r'_m$ such that if, for an $m$-layer continuum transformer $\mathcal{T} := \mathcal{T}_m \circ \ldots \circ \mathcal{T}_0$, $[\widetilde{H}(U, W)]_{i,j} = \kappa(u^{(i)}, w^{(j)})$, $\mathcal{W}_{v,\ell} = \begin{pmatrix} 0 & 0 \\ 0 & -r'_\ell I \end{pmatrix}$, $\mathcal{W}_{q,\ell} = I$, and $\mathcal{W}_{k,\ell} = I$ for each $\ell = 0, \ldots, m$, then for any $f \in \mathcal{X}$,*

$$\mathcal{T}_\ell(f; (\mathcal{W}_v, \mathcal{W}_q, \mathcal{W}_k)|z^{(1)}, \ldots, z^{(n)}) = -O_\ell f.$$

*Proof.* From calculations in subsection Appendix D.1, we know that the $\ell$-th step of gradient descent has the form

$$O_{\ell+1} = O_\ell + r'_\ell \sum_{i=1}^n \kappa(f^{(i)}, \cdot)(u^{(i)} - O_\ell f^{(i)}).$$

From the dynamics of the transformer, it can be easily shown by induction that $X_\ell \equiv X_0$ for all $\ell$.

We now prove that

$$u_\ell^{(j)} = u^{(j)} + \mathcal{T}_\ell(f^{(j)}; (\mathcal{W}_q, \mathcal{W}_v, \mathcal{W}_k)|z^{(1)}, \ldots, z^{(n)}) \tag{29}$$

for all $j = 0, \ldots, n$. In other words, "$u^{(j)} - u_\ell^{(j)}$ is equal to the predicted label for $f$, if $f^{(j)} = f$".

We prove this by induction. Let's explicitly write the dynamics for layer 1.

$$
\begin{aligned}
Z_1 &= Z_0 + \left( \widetilde{H}(\mathcal{W}_{q,0}X_0, \mathcal{W}_{k,0}X_0)\mathcal{M}(\mathcal{W}_{v,0}Z_0)^T \right)^T \\
&= Z_0 + \left( \widetilde{H}(X_0, X_0) \begin{bmatrix} I & 0 \\ 0 & 0 \end{bmatrix} \left( \begin{bmatrix} 0 & 0 \\ 0 & -r_0'I \end{bmatrix} \begin{bmatrix} f^{(1)} & \cdots & f^{(n)} & f \\ u^{(1)} & \cdots & u^{(n)} & u^{(n+1)} \end{bmatrix} \right)^T \right)^T \\
&= Z_0 + \left( \widetilde{H}(X_0, X_0) \begin{bmatrix} 0 & \cdots & 0 & 0 \\ -r_0'u^{(1)} & \cdots & -r_0'u^{(n)} & 0 \end{bmatrix}^T \right)^T \\
&= Z_0 + \left( \begin{bmatrix} \kappa(f^{(1)}, f^{(1)}) & \kappa(f^{(1)}, f^{(2)}) & \cdots & \kappa(f^{(1)}, f^{(n)}) & \kappa(f^{(1)}, f) \\ \kappa(f^{(2)}, f^{(1)}) & & \cdots & \kappa(f^{(2)}, f^{(n)}) & \kappa(f^{(2)}, f) \\ \vdots & & & \vdots & \\ \kappa(f, f^{(1)}) & & \cdots & \kappa(f, f^{(n)}) & \kappa(f, f) \end{bmatrix} \begin{bmatrix} 0 & -r_0'u^{(1)} \\ \vdots & \vdots \\ 0 & -r_0'u^{(n)} \\ 0 & 0 \end{bmatrix} \right)^T \\
&= Z_0 - r_0' \begin{bmatrix} 0 & \cdots & 0 & 0 \\ \sum_{i=1}^n \kappa(f^{(1)}, f^{(i)})u^{(i)} & \cdots & \sum_{i=1}^n \kappa(f^{(n)}, f^{(i)})u^{(i)} & \sum_{i=1}^n \kappa(f, f^{(i)})u^{(i)} \end{bmatrix} \\
&= \begin{bmatrix} f^{(1)} & \cdots & f^{(n)} & f \\ u^{(1)} - r_0'\sum_{i=1}^n \kappa(f^{(1)}, f^{(i)})u^{(i)} & \cdots & u^{(n)} - r_0'\sum_{i=1}^n \kappa(f^{(n)}, f^{(i)})u^{(i)} & -r_0'\sum_{i=1}^n \kappa(f, f^{(i)})u^{(i)} \end{bmatrix}.
\end{aligned}
$$

By definition, $\mathcal{T}_1(f; (\mathcal{W}_q, \mathcal{W}_v, \mathcal{W}_k)|z^{(1)}, \ldots, z^{(n)}) = -r_0'\sum_{i=1}^n \kappa(f, f^{(i)})u^{(i)}$. if we plug in $x = x^{(j)}$ for any column $j$, we recover the $j$-th column value in the bottom right. In other words, for the 1-layer case, we have

$$
u_1^{(j)} = u^{(j)} + \mathcal{T}_1(f^{(j)}; (\mathcal{W}_q, \mathcal{W}_v, \mathcal{W}_k)|z^{(1)}, \ldots, z^{(n)}).
$$

For the rest of the proof, we use $\mathcal{T}_\ell(f)$ to denote $\mathcal{T}_\ell(f); (\mathcal{W}_q, \mathcal{W}_v, \mathcal{W}_k)|z^{(1)}, \ldots, z^{(n)})$ to simplify notation. We now prove the induction case. Suppose that $u_\ell^{(i)} = u^{(i)} + \mathcal{T}_\ell(f^{(i)})$. Then, by exactly the same calculation above, for the $\ell + 1$ layer:

$$
\begin{aligned}
Z_{\ell+1} &= Z_\ell + \left( \widetilde{H}(\mathcal{W}_{q,\ell}X_\ell, \mathcal{W}_{k,\ell}X_\ell)\mathcal{M}(\mathcal{W}_{v,\ell}Z_\ell)^T \right)^T \\
&= Z_\ell + \left( \widetilde{H}(X_0, X_0)\mathcal{M}(\mathcal{W}_{v,\ell}Z_\ell)^T \right)^T \\
&= Z_\ell - r_\ell' \begin{bmatrix} 0 & \cdots & 0 & 0 \\ \sum_{i=1}^n \kappa(f^{(1)}, f^{(i)})u_\ell^{(i)} & \cdots & \sum_{i=1}^n \kappa(f^{(n)}, f^{(i)})u_\ell^{(i)} & \sum_{i=1}^n \kappa(f, f^{(i)})u_\ell^{(i)} \end{bmatrix}
\end{aligned}
\tag{30}
$$

where the second equation follows from the fact that $\mathcal{W}_{q,\ell}$ and $\mathcal{W}_{v,\ell}$ are identity operators and $X_\ell = X_0$. We now apply the induction hypothesis to the right hand side and get

$$
\begin{bmatrix} f^{(1)} & u^{(1)} + \mathcal{T}_\ell(f^{(1)}) - r_\ell'\sum_{i=1}^n \kappa(f^{(1)}, f^{(i)})(u^{(i)} + \mathcal{T}_\ell(f^{(i)})) \\ \vdots & \vdots \\ f^{(n)} & u^{(n)} + \mathcal{T}_\ell(f^{(n)}) - r_\ell'\sum_{i=1}^n \kappa(f^{(n)}, f^{(i)})(u^{(i)} + \mathcal{T}_\ell(f^{(i)})) \\ f & \mathcal{T}_\ell(f) - r_\ell'\sum_{i=1}^n \kappa(f, f^{(i)})(u^{(i)} + T F_\ell(f^{(i)})) \end{bmatrix}^T.
$$

Now $\mathcal{T}_{\ell+1}(f) = \mathcal{T}_\ell(f) - r_\ell'\sum_{i=1}^n \kappa(f, f^{(i)})(u^{(i)} + \mathcal{T}_\ell(f^{(i)}))$. Substituting $f^{(j)}$ in place of $f$ gives us

$$
u_{\ell+1}^{(j)} = u^{(j)} + \mathcal{T}_{\ell+1}(f^{(j)}).
$$

We now proceed to the proof of the theorem using induction. At step 0, $Z_0 := 0 = O_0$. Now assume $\mathcal{T}_\ell(f; (\mathcal{W}_q, \mathcal{W}_v, \mathcal{W}_k)|z^{(1)}, \ldots, z^{(n)}) = -O_\ell f$ holds up to some layer $\ell$. For the next layer $\ell + 1$,

$$
\begin{aligned}
\mathcal{T}_{\ell+1}(f; (\mathcal{W}_q, \mathcal{W}_v, \mathcal{W}_k)) &= \mathcal{T}_\ell(f; (\mathcal{W}_q, \mathcal{W}_v, \mathcal{W}_k)|z^{(1)}, \ldots, z^{(n)}) - r'_\ell \sum_{i=1}^n [\widetilde{H}(X_0, X_0)]_{n+1,i} u_\ell^{(i)} \\
&= \mathcal{T}_\ell(f; (\mathcal{W}_q, \mathcal{W}_v, \mathcal{W}_k)|z^{(1)}, \ldots, z^{(n)}) - r'_\ell \sum_{i=1}^n [\widetilde{H}(X_0, X_0)]_{n+1,i} (u^{(i)} - O_\ell f) \\
&= -O_\ell f - r'_\ell \sum_{i=1}^n \kappa(f, f^{(i)})(u^{(i)} - O_\ell f) \\
&= -O_{\ell+1} f.
\end{aligned}
$$

Here, the first line follows from plugging in $\mathcal{W}_q, \mathcal{W}_v, \mathcal{W}_k$ in Equation (30). The second line follows from Equation (29) and the induction hypothesis. □

# F  BEST LINEAR UNBIASED PREDICTOR

## F.1  BLUP COINCIDES WITH BAYES OPTIMAL

We now provide the formal statement that the Best Linear Unbiased Predictor (BLUP) and Bayes Optimal predictors coincide in this Hilbert space kriging setting of interest.

**Theorem F.1.** *(Theorem 4 of Menafoglio & Petris (2016)) Let $X_{n+1} \in \mathcal{X}$ and $\mathbf{X} = (X_1, \ldots, X_n) \in \mathcal{X}^n$ be zero-mean jointly Gaussian random variables. Assume that the covariance operator $C_\mathbf{X}$ is invertible. Then*

$$\mathbb{E}(X_{n+1}|\mathbf{X}) = L\,\mathbf{X},$$

*where the $L$ is the linear operator given by $L = C_{X_{n+1}\mathbf{X}} C_\mathbf{X}^{-1}$. Hence, the conditional expectation is an unbiased predictor and minimizes the mean squared prediction error*

$$\mathbb{E}[\|X_{n+1} - f(\mathbf{X})\|_\mathcal{X}^2]$$

*among all the measurable functions $f : \mathcal{X}^n \to \mathcal{X}$. The best predictor, in the mean squared norm sense, coincides with the Best Linear Unbiased Predictor.*

## F.2  IN-CONTEXT LEARNING BLUP PROOF

**Proposition F.2.** *Let $F = [f^{(1)}, \ldots, f^{(n+1)}], U = [u^{(1)}, \ldots, u^{(n+1)}]$. Let $\kappa : \mathcal{X} \times \mathcal{X} \to \mathcal{L}(\mathcal{X})$ be an operator-valued kernel. Assume that $U|F$ is a $\kappa$ Gaussian random variable per Definition 3.2. Let the activation function $\tilde{H}$ of the attention layer be defined as $[\tilde{H}(U, W)]_{i,j} := \kappa(u^{(i)}, w^{(j)})$. Consider the operator gradient descent in Theorem 3.1. Then as the number of layers $m \to \infty$, the continuum transformer's prediction at layer $m$ approaches the Best Linear Unbiased Predictor that minimizes the in-context loss in Equation (9).*

*Proof.* The notion of Best Linear Unbiased Predictor (BLUP) for a Gaussian random variable when all but one of the variables is observed, has been studied extensively in the literature on 'kriging'. This problem was solved for random variables in a Banach space in Luschgy (1996). Hilbert-space valued random variables would be a special case of this. This was dealt with in Menafoglio & Petris (2016). We use these results to find the BLUP for $u^{(n+1)}$ conditional on $u^{(1)}, \ldots, u^{(n)}$. First we partition the covariance operator $\mathbb{K}(F)$ into $\hat{\mathbb{K}}$, which represents the top-left $n \times n$ block. Let $\nu \in \mathcal{L}(\mathcal{X})^n$ denote the vector given by $\nu_i = \mathbb{K}_{i,n+1}$-

$$\mathbb{K} = \begin{bmatrix} \hat{\mathbb{K}} & \nu \\ \nu^T & \mathbb{K}_{n+1,n+1} \end{bmatrix}.$$

We note that $\hat{\mathbb{K}}$ is the covariance operator of the random variable $\hat{U} = [u^{(1)}, \ldots, u^{(n)}] \in \mathcal{X}^n$. The vector $\nu$ is the cross-covariance operator between the random variable $u^{(n+1)} \in \mathcal{X}$ and $\hat{U}$. Following Theorem 2 in Luschgy (1996), we assume that $\hat{\mathbb{K}}$ is injective. Then Theorem 3 in Menafoglio &

Petris (2016) (which is a Hilbert-space version of Theorem 2 in Luschgy (1996)) states that the Best Linear Unbiased Predictor with respect to the mean squared norm error

$$E(\|u^{(n+1)} - f(\hat{U})\|_{\mathcal{X}}^2)$$

among all measurable functions $f : \mathcal{X}^n \to \mathcal{X}$, is given by

$$\nu^T \hat{\mathbb{K}}^{-1} \hat{U} = \sum_{i,j=1}^n \kappa(f^{(n+1)}, f^{(i)})[\hat{\mathbb{K}}^{-1}]_{ij} u^{(j)}.$$

From the premise of 3.1, we know that $\mathcal{W}_{q,\ell} = I, \mathcal{W}_{k,\ell} = I, \mathcal{W}_{v,\ell} = \begin{bmatrix} 0 & 0 \\ 0 & -r'_\ell I \end{bmatrix}$. Set $r_\ell = \delta I$ for all $\ell$, where $\delta$ is a positive constant satisfying $\|I - \delta \hat{\mathbb{K}}\| < 1$, where the norm is the operator norm. From previous calculations done in Equation (30), we have:

$$u_{\ell+1}^{(i)} = u_\ell^{(i)} - \delta \sum_{j=1}^n \kappa(f^{(i)}, f^{(j)}) u_\ell^{(j)}.$$

We define the vector $\hat{U}_\ell := [u_\ell^{(1)}, \dots, u_\ell^{(n)}]$. Then in vector notation,

$$\hat{U}_{\ell+1} = (I - \delta \hat{\mathbb{K}}) \hat{U}_\ell.$$

Using induction on $\ell$ gives us:

$$\hat{U}_{\ell+1} = (I - \delta \hat{\mathbb{K}})^\ell \hat{U}.$$

Again, using Equation (30), we have:

$$u_{\ell+1}^{(n+1)} = u_\ell^{(n+1)} - \delta \sum_{j=1}^n \kappa(f^{(n+1)}, f^{(j)}) u_\ell^{(j)}.$$

In vector notation, this gives us

$$u_{\ell+1}^{(n+1)} = u_\ell^{(n+1)} - \delta \nu^T \hat{U}_\ell = -\delta \nu^T \sum_{k=0}^\ell \hat{U}_\ell = -\delta \nu^T \sum_{k=0}^\ell (I - \delta \hat{\mathbb{K}})^k \hat{U}.$$

Since $\delta$ was chosen such that $\|I - \delta \hat{\mathbb{K}}\| < 1$, $\hat{\mathbb{K}}^{-1} = \delta \sum_{k=0}^\infty (I - \delta \hat{\mathbb{K}})^k$. Hence, as $L \to \infty$, $u_{\ell+1}^{(n+1)} \to \nu^T \hat{\mathbb{K}}^{-1} \hat{U}$, which is the BLUP of $u^{(n+1)}$. □

## G  OPTIMIZATION CONVERGENCE PROOF

We begin by reviewing some basic notions of differentiation in Hilbert spaces, for which we follow Clément (2009).

**Definition G.1.** Let $L : \mathcal{X} \to \mathbb{R}$ be a functional from a Hilbert space $\mathcal{X}$ to real numbers. We say that $L$ is Fréchet-differentiable at $f_0 \in \mathcal{X}$ if there exists a bounded linear operator $A : \mathcal{X} \to \mathbb{R}$ and a function $\phi : \mathbb{R} \to \mathbb{R}$ with $\lim_{s \to 0} \frac{\phi(s)}{s} = 0$ such that

$$L(f_0 + h) - L(f_0) = Ah + \phi(\|h\|_{\mathcal{X}}).$$

In this case we define $DL(f_0) = A$.

**Definition G.2.** If $L$ is Fréchet-differentiable at $f_0 \in \mathcal{X}$ with $DL(f_0) \in \mathcal{X}$, then the $\mathcal{X}$-gradient $\nabla_{\mathcal{X}} L(f_0) \in \mathcal{X}$ is defined by

$$\langle \nabla_{\mathcal{X}} L(f_0), f \rangle = DL(f_0)(f)$$

for all $f \in \mathcal{X}$.

We now state the assumptions required to prove the results of the optimization landscape.

**Assumption G.3.** (Rotational symmetry of distributions) Let $P_F$ denote the distribution of $F = [f^{(1)}, \ldots, f^{(n+1)}]$ and $\mathbb{K}(F) = \mathbb{E}_{U|F}[U \otimes U]$. We assume that there exists a self-adjoint, invertible operator $\Sigma : \mathcal{X} \to \mathcal{X}$ such that for any unitary operator $\mathcal{M}$, $\Sigma^{1/2} \mathcal{M} \Sigma^{-1/2} F \overset{d}{=} F$ and $\mathbb{K}(F) = \mathbb{K}(\Sigma^{1/2} \mathcal{M} \Sigma^{-1/2} F)$.

**Assumption G.4.** For any $F_1, F_2 \in \mathcal{X}^{n+1}$ and any operator $S : \mathcal{X} \to \mathcal{X}$ with an inverse $S^{-1}$, the function $\tilde{H}$ satisfies $\tilde{H}(F_1, F_2) = \tilde{H}(S^* F_1, S^{-1} F_2)$.

**Theorem G.5.** *Suppose Assumption G.3 and Assumption G.4 hold. Let $f(r, \mathcal{W}_q, \mathcal{W}_k) := L\left( \mathcal{W}_{v,\ell} = \left\{ \begin{bmatrix} 0 & 0 \\ 0 & r_\ell \end{bmatrix} \right\}_{\ell=0,\ldots,m}, \mathcal{W}_{q,\ell}, \mathcal{W}_{k,\ell} \right)$, where $L$ is as defined in Equation* (9). *Let $\mathcal{S} \subset \mathcal{O}^{m+1} \times \mathcal{O}^{m+1}$ denote the set of $(\mathcal{W}_q, \mathcal{W}_k)$ operators with the property that $(\mathcal{W}_q, \mathcal{W}_v) \in \mathcal{S}$ if and only if for all $\ell \in \{0, \ldots, m\}$, there exist scalars $b_\ell, c_\ell \in \mathbb{R}$ such that $\mathcal{W}_{q,\ell} = b_\ell \Sigma^{-1/2}$ and $\mathcal{W}_{k,\ell} = c_\ell \Sigma^{-1/2}$. Then*

$$\inf_{(r, (\mathcal{W}_q, \mathcal{W}_k)) \in \mathbb{R}^{m+1} \times \mathcal{S}} \sum_{\ell=0}^{m} \left[ (\partial_{r_\ell} f)^2 + \|\nabla_{\mathcal{W}_{q,\ell}} f\|_{\mathrm{HS}}^2 + \|\nabla_{\mathcal{W}_{k,\ell}} f\|_{\mathrm{HS}}^2 \right] = 0. \tag{31}$$

*Here $\nabla_{\mathcal{W}_{q,\ell}}$ and $\nabla_{\mathcal{W}_{k,\ell}}$ denote derivatives with respect to the Hilbert-Schmidt norm $\|\cdot\|_{\mathrm{HS}}$.*

The key insight to generalizing the proof for functional gradient descent in Cheng et al. (2023) is that in the finite-dimensional case, we are working in the Hilbert space $\mathbb{R}^{d \times d}$, equipped with the Frobenius norm. In the current setting of operators, we work with an arbitrary Hilbert space. We refer the reader to Clément (2009) for discussion on the existence and uniqueness of gradient flows in Hilbert spaces.

*Proof.* We define $\mathcal{S}$-gradient flows as

$$\frac{d}{dt} r_\ell(t) = \partial_{r_\ell} L(r(t), \mathcal{W}_q(t), \mathcal{W}_k(t))$$

$$\frac{d}{dt} \mathcal{W}_{q,\ell}(t) = \tilde{B}_\ell(t)$$

$$\frac{d}{dt} \mathcal{W}_{k,\ell}(t) = \tilde{C}_\ell(t)$$

where

$$b_\ell(t) := \langle \nabla_{\mathcal{W}_{q,\ell}} L(r(t), \mathcal{W}_q(t), \mathcal{W}_k(t)), \Sigma^{1/2} \rangle \quad \tilde{B}_\ell(t) := b_\ell(t) \Sigma^{-1/2}$$

$$c_\ell(t) := \langle \nabla_{\mathcal{W}_{k,\ell}} L(r(t), \mathcal{W}_q(t), \mathcal{W}_k(t)), \Sigma^{1/2} \rangle \quad \tilde{C}_\ell(t) := c_\ell(t) \Sigma^{-1/2}.$$

We observe that the definition of the functions $B$ and $C$ ensures that $\mathcal{W}_{q,\ell}(t), \mathcal{W}_{k,\ell}(t) \in \mathcal{S}$ for all $t$.

$$\frac{d}{dt} L(r(t), \mathcal{W}_q(t), \mathcal{W}_k(t))$$

$$= \sum_{\ell=0}^{k} \partial_{r_\ell} L(r(t), \mathcal{W}_q(t), \mathcal{W}_k(t)) \cdot (-\partial_{r_\ell} L(r(t), \mathcal{W}_q(t), \mathcal{W}_k(t))) \tag{32}$$

$$+ \sum_{\ell=0}^{k} \langle \nabla_{\mathcal{W}_{q,\ell}} L(r(t), \mathcal{W}_q(t), \mathcal{W}_k(t)), \tilde{B}_\ell(t) \rangle_{HS} \tag{33}$$

$$+ \sum_{\ell=0}^{k} \langle \nabla_{\mathcal{W}_{k,\ell}} L(r(t), \mathcal{W}_q(t), \mathcal{W}_k(t)), \tilde{C}_\ell(t) \rangle_{HS}. \tag{34}$$

Clearly, Equation (32) $= -\sum_{\ell=0}^{k}(\partial_{r_\ell} L(r(t), \mathcal{W}_q(t), \mathcal{W}_k(t)))^2$. From Proposition G.6, we note that

$$(33) \leq \sum_{\ell=0}^{k} \langle \nabla_{\mathcal{W}_{q,\ell}} L(r(t), \mathcal{W}_q(t), \mathcal{W}_k(t)), -\nabla_{\mathcal{W}_{q,\ell}} L(r(t), \mathcal{W}_q(t), \mathcal{W}_k(t)) \rangle$$

$$= -\sum_{\ell=0}^{k} \|\nabla_{\mathcal{W}_{q,\ell}} L(r(t), \mathcal{W}_q(t), \mathcal{W}_k(t))\|_{HS}^2.$$

Similarly,

$$(34) \leq -\sum_{\ell=0}^{k} \|\nabla_{\mathcal{W}_{k,\ell}} L(r(t), \mathcal{W}_q(t), \mathcal{W}_k(t))\|_{HS}^2.$$

We have shown that at any time $t$,

$$\frac{d}{dt} L(r(t), \mathcal{W}_q(t), \mathcal{W}_k(t)) \leq -\sum_{\ell=0}^{k}(\partial_{r_\ell} L(r(t), \mathcal{W}_q(t), \mathcal{W}_k(t)))^2$$

$$-\sum_{\ell=0}^{k} \|\nabla_{\mathcal{W}_{q,\ell}} L(r(t), \mathcal{W}_q(t), \mathcal{W}_k(t))\|_{HS}^2 - \sum_{\ell=0}^{k} \|\nabla_{\mathcal{W}_{k,\ell}} L(r(t), \mathcal{W}_q(t), \mathcal{W}_k(t))\|_{HS}^2$$

Suppose Equation (12) does not hold. Then there exists a positive constant $c > 0$ such that for all $t$,

$$\sum_{\ell=0}^{k}(\partial_{r_\ell} L(r(t), \mathcal{W}_q(t), \mathcal{W}_k(t)))^2 + \sum_{\ell=0}^{k} \|\nabla_{\mathcal{W}_{q,\ell}} L(r(t), \mathcal{W}_q(t), \mathcal{W}_k(t))\|_{HS}^2$$

$$+ \sum_{\ell=0}^{k} \|\nabla_{\mathcal{W}_{k,\ell}} L(r(t), \mathcal{W}_q(t), \mathcal{W}_k(t))\|_{HS}^2 \geq c.$$

This implies that $\frac{d}{dt} L(r(t), \mathcal{W}_q(t), \mathcal{W}_k(t)) \leq -c$ for all $t$, which is a contradiction since $L(\cdot)$ is bounded below by zero. Hence, we have proved that Equation (12) holds.

$\square$

**Proposition G.6.** *Suppose $F, U$ satisfy Assumption G.3 and $\tilde{H}$ satisfies Assumption G.4. Suppose $\mathcal{W}_q, \mathcal{W}_k \in \mathcal{S}$. Fix a layer $j \in \{0, \dots, m\}$. For any $R \in \mathcal{L}(\mathcal{X})$, let $\mathcal{W}_q(R, j)$ denote the collection of operators where $[\mathcal{W}_q(R, j)]_j := \mathcal{W}_{q,j} + R$ and $[\mathcal{W}_q(R, j)]_\ell = \mathcal{W}_{q,\ell}$ for all $\ell \neq j$. Take an arbitrary $R \in \mathcal{L}(\mathcal{X})$. Let*

$$\tilde{R} = \frac{1}{d} Tr(R\Sigma^{1/2})\Sigma^{-1/2},$$

*where $R\Sigma^{1/2}$ denotes composition of operators and $Tr$ is the trace of an operator as defined in Definition G.10. Then for any $j \in \{0, \dots, m\}$,*

$$\frac{d}{dt} L(r, \mathcal{W}_q(tR, j), \mathcal{W}_k)\Big|_{t=0} \leq \frac{d}{dt} L(r, \mathcal{W}_q(t\tilde{R}, j), \mathcal{W}_k)\Big|_{t=0} \tag{35}$$

*and*

$$\frac{d}{dt} L(r, \mathcal{W}_q, \mathcal{W}_k(tR, j))\Big|_{t=0} \leq \frac{d}{dt} L(r, \mathcal{W}_q, \mathcal{W}_k(t\tilde{R}, j))\Big|_{t=0}. \tag{36}$$

Since the proofs of Equation (35) and Equation (36) are similar, we only present the proof of Equation (35).

Suppose $\mathcal{W}_q(R)$ be the collection of operators where $[\mathcal{W}_q(R)]_\ell = \mathcal{W}_{q,\ell} + R$ for all $\ell \in \{0, \dots, m\}$. Since Proposition G.6 holds for any $j$, and

$$\frac{d}{dt} L(r, \mathcal{W}_q(tR), \mathcal{W}_k) = \sum_{j=0}^{m} \frac{d}{dt} L(r, \mathcal{W}_q(tR, j), \mathcal{W}_k),$$

an immediate consequence of Proposition G.6 is that

$$\frac{d}{dt}L(r, \mathcal{W}_q(tR), \mathcal{W}_k)\Big|_{t=0} \leq \frac{d}{dt}L(r, \mathcal{W}_q(t\tilde{R}), \mathcal{W}_k)\Big|_{t=0}.$$

The rest of this section is dedicated to proving this proposition. As a first step, we re-write the in-context loss using an inner product.

**Lemma G.7.** *(**Re-writing the In-Context Loss**) Let $\bar{Z}_0$ be the input of the transformer where the last entry of $Y$ has not been masked (or zeroed out):*

$$\bar{Z}_0 = \begin{bmatrix} f^{(1)} & \cdots & f^{(n)} & f^{(n+1)} \\ u^{(1)} & \cdots & u^{(n)} & u^{(n+1)} \end{bmatrix}.$$

*Let $\bar{Z}_\ell$ be the output of the $(\ell-1)^{th}$ layer of the transformer initialized at $\bar{Z}_0$. Let $\bar{F}_\ell$ and $\bar{U}_\ell$ denote the first and second row of $\bar{Z}_\ell$. Then the in-context loss defined in Equation (9) has the equivalent form*

$$L(\mathcal{W}_v, \mathcal{W}_q, \mathcal{W}_k) = E_{\bar{Z}_0}[\langle (I-M)\bar{U}_{m+1}^T, (I-M)\bar{U}_{m+1}^T \rangle].$$

*The Hilbert space corresponds to the direct sum of Hilbert spaces $\bigoplus_{i=1}^{n+1} \mathcal{X}$, where the inner product is given by*

$$\langle (v_1, \ldots, v_{n+1})^T, (w_1, \ldots, w_{n+1})^T \rangle = \sum_{i=1}^{n+1} \langle v_i, w_i \rangle_{\mathcal{X}}.$$

*Proof.* We deviate from the mathematical convention where an operator that acts on a function is always written to the left of the function. We adopt the convention that $fO$, where $O$ is an operator and $f$ is a function, also means that the operator acts on the function. This is how the equations henceforth should be interpreted.

When $\mathcal{W}_{v,\ell} = \begin{bmatrix} A_\ell & 0 \\ 0 & r_\ell I \end{bmatrix}$, the output at each layer, given in Equation (7) can be simplified as follows:

$$\begin{bmatrix} F_{\ell+1} \\ U_{\ell+1} \end{bmatrix} = \begin{bmatrix} F_\ell \\ U_\ell \end{bmatrix} + \left( \tilde{H}(\mathcal{W}_{q,\ell}F_\ell, \mathcal{W}_{k,\ell}F_\ell)M \left( \begin{bmatrix} A_\ell & 0 \\ 0 & r_\ell I \end{bmatrix} \begin{bmatrix} F_\ell & f_\ell^{(n+1)} \\ U_\ell & 0 \end{bmatrix} \right)^T \right)^T$$

$$= \begin{bmatrix} F_\ell \\ U_\ell \end{bmatrix} + \left( \tilde{H}(\mathcal{W}_{q,\ell}F_\ell, \mathcal{W}_{k,\ell}F_\ell) \begin{bmatrix} I & 0 \\ 0 & 0 \end{bmatrix} \left( \begin{bmatrix} A_\ell F_\ell & A_\ell f_\ell^{(n+1)} \\ r_\ell U_\ell & 0 \end{bmatrix} \right)^T \right)^T$$

$$= \begin{bmatrix} F_\ell \\ U_\ell \end{bmatrix} + \left( \begin{bmatrix} \kappa(f_\ell^{(1)}, f_\ell^{(1)}) & \cdots & \kappa(f_\ell^{(1)}, f_\ell^{(n)}) & \kappa(f_\ell^{(1)}, f_\ell) \\ \kappa(f_\ell^{(2)}, f_\ell^{(1)}) & \cdots & \kappa(f_\ell^{(2)}, f_\ell^{(n)}) & \kappa(f_\ell^{(2)}, f_\ell) \\ \vdots & & \vdots & \vdots \\ \kappa(f_\ell, f_\ell^{(1)}) & \cdots & \kappa(f_\ell, f_\ell^{(n)}) & \kappa(f_\ell, f_\ell) \end{bmatrix} \begin{bmatrix} (A_\ell F_\ell)^T & (r_\ell U_\ell)^T \\ 0 & 0 \end{bmatrix} \right)^T$$

$$= \begin{bmatrix} F_\ell \\ U_\ell \end{bmatrix} + \begin{bmatrix} A_\ell F_\ell M \tilde{H}(\mathcal{W}_{q,\ell}F_\ell, \mathcal{W}_{k,\ell}F_\ell) \\ r_\ell U_\ell M \tilde{H}(\mathcal{W}_{q,\ell}F_\ell, \mathcal{W}_{k,\ell}F_\ell) \end{bmatrix}. \tag{37}$$

Suppose $\bar{Z}_0$ is equal to $Z_0$ everywhere, except the $(2, n+1)^{th}$ entry, where it is $c$ instead of $0$. We compare the dynamics of $Z_\ell$ and $\bar{Z}_\ell$. Since $F_0 = \bar{F}_0$, we see that $F_\ell = \bar{F}_\ell$. We claim that $\bar{U}_\ell - U_\ell = [0, \ldots, 0, c]$. We prove this by induction. This is trivially true for $\ell = 0$. Suppose this holds for step $\ell$, then at step $\ell+1$:

$$\bar{U}_{\ell+1} = \bar{U}_\ell + r_\ell \bar{U}_\ell M \tilde{H}(\mathcal{W}_{q,\ell}\bar{F}_\ell, \mathcal{W}_{k,\ell}\bar{F}_\ell)$$

$$= \bar{U}_\ell + r_\ell U_\ell M \tilde{H}(\mathcal{W}_{q,\ell}F_\ell, \mathcal{W}_{k,\ell}F_\ell)$$

$$= U_\ell + [0, \ldots, 0, c] + r_\ell U_\ell M \tilde{H}(\mathcal{W}_{q,\ell}F_\ell, \mathcal{W}_{k,\ell}F_\ell)$$

$$= U_{\ell+1} + [0, \ldots, 0, c],$$

where the second equality follows from the fact that $\bar{U}_\ell M$ has its $(n+1)^{th}$ entry zeroed out, so by the inductive hypothesis, $\bar{U}_\ell M = U_\ell M$. The third equality follows from the inductive hypothesis again.

Replacing $c$ by $u^{(n+1)}$ tells us that $[\bar{Z}_{m+1}]_{2,n+1} = [Z_{m+1}]_{2,n+1} + u^{(n+1)}$. Hence, the loss in Equation (9) can be re-written as

$$
\begin{aligned}
L(\mathcal{W}_v, \mathcal{W}_q, \mathcal{W}_k) &= \mathbb{E}_{\bar{Z}_0}\left[\left\|[\bar{Z}_{m+1}]_{2,n+1}\right\|^2\right] \\
&= \mathbb{E}_{\bar{Z}_0}\left[\left\|\bar{U}_{m+1}(I - M)\right\|^2\right] \\
&= \mathbb{E}_{\bar{Z}_0}\left[\langle(I-M)\bar{U}_{m+1}^T, (I-M)\bar{U}_{m+1}^T\rangle\right],
\end{aligned}
\tag{38}
$$

where the inner product in the last line is the inner product on column vectors in $\bigoplus_{i=1}^{n+1}\mathcal{X}$. $\square$

Since $A_\ell = 0$, an immediate corollary of Equation (37) is as follows.

**Corollary G.8.** *When $A_\ell = 0$ for all $\ell \in \{0, \ldots, m\}$,*
$$
F_{\ell+1} \equiv F_0.
$$
*Moreover,*
$$
\begin{aligned}
U_{\ell+1} &= U_\ell + r_\ell U_\ell M \tilde{H}(\mathcal{W}_{q,\ell}F_0, \mathcal{W}_{k,\ell}F_0) \\
&= U_0 \prod_{i=0}^{\ell}(I + r_i M \tilde{H}(\mathcal{W}_{q,i}F_0, \mathcal{W}_{k,i}F_0)).
\end{aligned}
$$

We now define the trace operator and the tensor product, which allows us to use the covariance operator.

**Definition G.9.** Let $x_1, x_2$ be elements of the Hilbert spaces $\mathcal{X}_1$ and $\mathcal{X}_2$. The tensor product operator $(x_1 \otimes x_2) : \mathcal{X}_1 \to \mathcal{X}_2$ is defined by:
$$
(x_1 \otimes x_2)x = \langle x_1, x\rangle x_2
$$
for any $x \in \mathcal{X}_1$.

**Definition G.10.** Let $\mathcal{X}$ be a separable Hilbert space and let $\{e_i\}_{i=1}^{\infty}$ be a complete orthonormal system (CONS) in $\mathcal{X}$. If $T \in L_1(\mathcal{X}, \mathcal{X})$ be a nuclear operator. Then we define
$$
Tr\, T = \sum_{i=1}^{\infty}\langle Te_i, e_i\rangle.
$$

Henceforth, we assume that our Hilbert space is separable.

**Lemma G.11.** *Suppose $x_1, x_2 \in \mathcal{X}$, be used to construct the tensor product $x_1 \otimes x_2$. Then*
$$
Tr\, x_1 \otimes x_2 = \langle x_1, x_2\rangle.
$$

*Proof.* Let $\{e_i\}_{i=1}^{\infty}$ be a CONS for the Hilbert space $\mathcal{X}$. The inner product can be written as
$$
\begin{aligned}
\langle x_1, x_2\rangle &= \left\langle \sum_{i=1}^{\infty}\langle x_1, e_i\rangle e_i, \sum_{i=1}^{\infty}\langle x_2, e_i\rangle e_i, \right\rangle \\
&= \sum_{i=1}^{\infty}\langle x_1, e_i\rangle\langle x_2, e_i\rangle.
\end{aligned}
$$
Similarly, the trace of the tensor product is given by
$$
\begin{aligned}
Tr\, x_1 \otimes x_2 &= \sum_{i=1}^{\infty}\langle(x_1 \otimes x_2)e_i, e_i\rangle \\
&= \sum_{i=1}^{\infty}\langle\langle x_1, e_i\rangle x_2, e_i\rangle \\
&= \sum_{i=1}^{\infty}\langle x_1, e_i\rangle\langle x_2, e_i\rangle.
\end{aligned}
$$
This completes the proof of the lemma. $\square$

We also state some properties of these operators without proof:

1. Let $A$ be a linear operator from $\mathcal{X} \to \mathcal{X}$ and $x_1 \in \mathcal{X}$. Then

$$(Ax_1 \otimes Ax_1) = A(x_1 \otimes x_1)A^*,$$

where $A^*$ is the adjoint of $A$. This can be verified by writing out how each of the above operators acts on an arbitrary element $x \in \mathcal{X}$.

2. The cyclic property of trace, that is, $Tr(AB) = Tr(BA)$ also holds in infinite-dimensional spaces if $A, B$ are both Hilbert-Schmidt operators. This can be verified using a CONS.

The rest of the proof for Proposition G.6 has the following outline:

1. We reformulate the in-context loss as the expectation of a trace operator.
2. We introduce a function $\phi : \mathcal{X}^{n+1} \times \mathcal{O} \to \mathcal{O}^{(n+1)\times(n+1)}$ which is used to simplify the loss equation.
3. We understand the dynamics of $\phi$ over time.
4. We use all the identities we have proved to complete the proof.

*In-Context Loss as the Expectation of a Trace-*
Using Lemma G.11, we can write the in-context loss in Equation (38) as

$$L(\mathcal{W}_v, \mathcal{W}_q, \mathcal{W}_k) = \mathbb{E}_{\bar{Z}_0} \left[ Tr[((I-M)\bar{U}_{m+1}^T) \otimes ((I-M)\bar{U}_{m+1}^T)] \right]$$

Since $(I - M)$ is a self-adjoint operator,

$$((I-M)\bar{U}_{m+1}^T) \otimes ((I-M)\bar{U}_{m+1}^T) = (I-M)(\bar{U}_{m+1}^T \otimes \bar{U}_{m+1}^T)(I-M).$$

Then

$$L(\mathcal{W}_v, \mathcal{W}_q, \mathcal{W}_k) = \mathbb{E}_{\bar{Z}_0} \left[ Tr[(I-M)(\bar{U}_{m+1}^T \otimes \bar{U}_{m+1}^T)(I-M)] \right].$$

*Simplifying the Loss using the Function $\phi$-*
We drop the bar to simplify notation and denote $F_0$ by $F$. We also fix a $j \in \{0, \ldots, m\}$. We define the functional $\phi^j : \mathcal{X}^{n+1} \times \mathcal{O} \to \mathcal{O}^{(n+1)\times(n+1)}$ as

$$\phi^j(F, S) = \prod_{\ell=0}^{m}(I + r_\ell M \tilde{H}(\mathcal{W}_{q,\ell}(S, j)F, \mathcal{W}_{k,\ell}(S)F)).$$

Again, for the purpose of simplifying notation, we suppress the index $j$ since the proof follows through for any index. We use $\phi$ to denote $\phi^j$ and $\mathcal{W}_{q,\ell}(S)$ to denote $\mathcal{W}_{q,\ell}(S, j)$.

The loss can be reformulated as

$$\begin{aligned}
L(\mathcal{W}_v, \mathcal{W}_q, \mathcal{W}_k) &= \mathbb{E}_{Z_0} \left[ Tr[(I-M)((U_0\phi(F,S))^T \otimes (U_0\phi(F,S))^T)(I-M)] \right] \\
&= \mathbb{E}_{Z_0} \left[ Tr[(I-M)\phi^*(F,S)(U_0^T \otimes U_0^T)\phi(F,S)(I-M)] \right] \\
&= \mathbb{E}_{F_0} \left[ Tr[(I-M)\phi^*(F,S)\mathbb{K}(F_0)\phi(F,S)(I-M)] \right].
\end{aligned}$$

where the last equality follows from Assumption G.3 and the linearity and cyclic property of trace.

Let $\mathcal{U}$ be a uniformly randomly sampled unitary operator. Let $\mathcal{U}_\Sigma = \Sigma^{1/2}\mathcal{U}\Sigma^{-1/2}$. Using Assumption G.3, $\Sigma^{1/2}\mathcal{M}\Sigma^{-1/2}F \overset{d}{=} F$, we see that

$$\begin{aligned}
\frac{d}{dt}L(\mathcal{W}_v, \mathcal{W}_q(tR), \mathcal{W}_k)\Big|_{t=0} &= \frac{d}{dt}\mathbb{E}_{F_0}\left[Tr[(I-M)\phi^*(F_0, tR)\mathbb{K}(F_0)\phi(F_0, tR)(I-M)]\right]\Big|_{t=0} \\
&= 2\mathbb{E}_{F_0}\left[Tr[(I-M)\phi^*(F_0, tR)\mathbb{K}(F_0)\frac{d}{dt}\phi(F_0, 0)\Big|_{t=0}(I-M)]\right] \\
&= 2\mathbb{E}_{F_0, \mathcal{U}}\left[Tr[(I-M)\phi^*(\mathcal{U}_\Sigma F_0, 0)\mathbb{K}(F_0)\frac{d}{dt}\phi(\mathcal{U}_\Sigma F_0, tR)\Big|_{t=0}(I-M)]\right]
\end{aligned}$$

$$(39)$$

where the last equality uses the assumption $\mathbb{K}(\mathcal{U}_\Sigma F_0) = \mathbb{K}(F_0)$ from Assumption G.3.

*Dynamics of $\phi$ over Time-*
We now prove the following identities-

$$\phi(\mathcal{U}_\Sigma F_0, 0) = \phi(F_0, 0) \tag{40}$$

and

$$\frac{d}{dt}\phi(\mathcal{U}_\Sigma F_0, tR)\Big|_{t=0} = \frac{d}{dt}\phi(F_0, t\mathcal{U}_\Sigma^T R\mathcal{U}_\Sigma)\Big|_{t=0}. \tag{41}$$

We also recall that

$$\mathcal{W}_{q,\ell}\mathcal{U}_\Sigma = b_\ell \Sigma^{-1/2}\Sigma^{1/2}\mathcal{U}\Sigma^{-1/2} = \mathcal{U}\mathcal{W}_{q,\ell}. \tag{42}$$

Similarly,

$$\mathcal{W}_{k,\ell}\mathcal{U}_\Sigma = \mathcal{U}_\Sigma\mathcal{W}_{k,\ell}. \tag{43}$$

We now verify Equation (40).

$$\phi(\mathcal{U}_\Sigma F_0, 0) = \prod_{\ell=0}^m (I + r_\ell M\tilde{H}(\mathcal{W}_{q,\ell}\mathcal{U}_\Sigma(S)F_0, \mathcal{W}_{k,\ell}\mathcal{U}_\Sigma(S)F_0))$$

$$= \prod_{\ell=0}^m (I + r_\ell M\tilde{H}(\mathcal{U}_\Sigma\mathcal{W}_{q,\ell}(S)F_0, \mathcal{U}_\Sigma\mathcal{W}_{k,\ell}(S)F_0))$$

$$= \prod_{\ell=0}^m (I + r_\ell M\tilde{H}(\mathcal{W}_{q,\ell}(S)F_0, \mathcal{W}_{k,\ell}(S)F_0)) = \phi(F_0, 0),$$

where the last equality follows from the rotational invariance of $\tilde{H}$ from Assumption G.4.

We verify the following identity that will be used later on-

$$\frac{d}{dt}\tilde{H}((\mathcal{W}_{q,\ell} + tS)\mathcal{U}_\Sigma F_0, \mathcal{W}_{k,\ell}\mathcal{U}_\Sigma F_0)\Big|_{t=0} = \frac{d}{dt}\tilde{H}(\mathcal{U}\mathcal{W}_{q,\ell}F_0 + tS\mathcal{U}_\Sigma F_0, \mathcal{U}\mathcal{W}_{k,\ell}F_0)\Big|_{t=0}$$

$$= \frac{d}{dt}\tilde{H}(\mathcal{W}_{q,\ell}F_0 + t\mathcal{U}^T S\mathcal{U}_\Sigma F_0, \mathcal{W}_{k,\ell}F_0)\Big|_{t=0}, \tag{44}$$

where the first equality follows from Equation (42) and Equation (43) and the second equality follows from Assumption G.4.

Using the chain rule, we get

$$\frac{d}{dt}\phi(\mathcal{U}_\Sigma F_0, tR)\Big|_{t=0}$$

$$= \prod_{j=0}^m \left(\prod_{\ell=0}^{j-1}(I + M\tilde{H}(\mathcal{W}_{q,\ell}\mathcal{U}_\Sigma F_0, \mathcal{W}_{k,\ell}\mathcal{U}_\Sigma F_0))\right) M\frac{d}{dt}\tilde{H}((\mathcal{W}_{q,\ell} + tR)\mathcal{U}_\Sigma F_0, \mathcal{W}_{k,\ell}\mathcal{U}_\Sigma F_0)\Big|_{t=0}$$

$$\left(\prod_{\ell=j+1}^m (I + M\tilde{H}(\mathcal{W}_{q,\ell}\mathcal{U}_\Sigma F_0, \mathcal{W}_{k,\ell}\mathcal{U}_\Sigma F_0))\right)$$

$$= \prod_{j=0}^m \left(\prod_{\ell=0}^{j-1}(I + M\tilde{H}(\mathcal{W}_{q,\ell}F_0, \mathcal{W}_{k,\ell}F_0))\right) M\frac{d}{dt}\tilde{H}(\mathcal{W}_{q,\ell}F_0 + t\mathcal{U}^T R\mathcal{U}_\Sigma F_0, \mathcal{W}_{k,\ell}F_0)\Big|_{t=0}$$

$$\left(\prod_{\ell=j+1}^m (I + M\tilde{H}(\mathcal{W}_{q,\ell}F_0, \mathcal{W}_{k,\ell}F_0))\right)$$

$$= \frac{d}{dt}\phi(\mathcal{U}_\Sigma F_0, t\mathcal{U}^T R\mathcal{U}_\Sigma R)\Big|_{t=0},$$

where the second equality follows from Equation (42), Assumption G.4 and Equation (44).

*Putting it Together-*
Continuing from Equation (39), we get

$$\frac{d}{dt}L(\mathcal{W}_v, \mathcal{W}_q(tR), \mathcal{W}_k)\Big|_{t=0} = 2\mathbb{E}_{F_0, \mathcal{U}}\left[Tr[(I-M)\phi^*(\mathcal{U}_\Sigma F_0, 0)\mathbb{K}(F_0)\frac{d}{dt}\phi(\mathcal{U}_\Sigma F_0, t\mathcal{U}^T R\mathcal{U}_\Sigma)\Big|_{t=0}(I-M)]\right]$$

$$= 2\mathbb{E}_{F_0, \mathcal{U}}\left[Tr[(I-M)\phi^*(F_0, 0)\mathbb{K}(F_0)\frac{d}{dt}\phi(F_0, t\mathcal{U}^T R\mathcal{U}_\Sigma)\Big|_{t=0}(I-M)]\right]$$

$$= 2\mathbb{E}_{F_0}\left[Tr[(I-M)\phi^*(F_0, 0)\mathbb{K}(F_0)\frac{d}{dt}\phi(F_0, t\mathbb{E}_\mathcal{U}\left[\mathcal{U}^T R\mathcal{U}_\Sigma\right])\Big|_{t=0}(I-M)]\right]$$

$$= 2\mathbb{E}_{F_0}\left[Tr[(I-M)\phi^*(F_0, 0)\mathbb{K}(F_0)\frac{d}{dt}\phi(F_0, t\tilde{R})\Big|_{t=0}(I-M)]\right]$$

$$= \frac{d}{dt}L(\mathcal{W}_v, \mathcal{W}_q(t\tilde{R}), \mathcal{W}_k)\Big|_{t=0}.$$

Here, the first equality follows from Equation (41) and the second equality follows from Equation (40). The third equality uses the linearity of $\frac{d}{dt}\phi(F_0, tS)$ in $S$ and the fact that it is jointly continuously differentiable. This concludes the proof.

## H    EXPERIMENT KERNEL DEFINITIONS

We provide below the explicit definitions of the kernels studied in the experiments of Section 5.1:

**Kernels for $k_x(f, f')$**

| Name | Definition |
|------|------------|
| Linear | $\langle f, f'\rangle_\mathcal{X}$ |
| Laplacian | $\exp\left(-\frac{\|f-f'\|_\mathcal{X}}{\sigma_x}\right)$ |
| Gradient RBF | $\exp\left(-\frac{\|\nabla f - \nabla f'\|_\mathcal{X}^2}{2\sigma_x^2}\right)$ |
| Energy | $\exp\left(-\frac{(\|f\|_\mathcal{X}^2 - \|f'\|_\mathcal{X}^2)^2}{2\sigma_x^2}\right)$ |

**Kernels for $k_y(x, y)$**

| Name | Definition |
|------|------------|
| Laplace | $\exp\left(-\frac{\|x-y\|_2}{\sigma_y}\right)$ |
| Gaussian | $\exp\left(-\frac{\|x-y\|_2^2}{2\sigma_y^2}\right)$ |

# I   BLUP ADDITIONAL EXPERIMENTAL RESULTS

We present below the in-context learning predictions for each pair of the true data-generating kernel and choice of $k_x$ kernel, in the below visualizations. Paralleling the results seen in Figure 1, we find that, when $k_x$ matches the data-generating kernel, the resulting field prediction structurally matches the true $\mathcal{O}f$. Across these visualizations, we fix $k_y$ to be the Gaussian for simplicity of presentation.

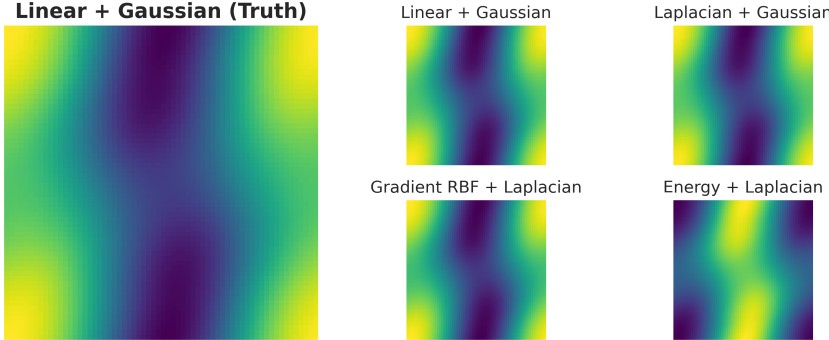

Figure 4: In-context predictions for $(k_x, k_y)$ being (Linear, Gaussian), (Laplacian, Gaussian), (Gradient RBF, Laplace), and (Energy, Laplace) with the data-generating kernel being (Linear, Gaussian).

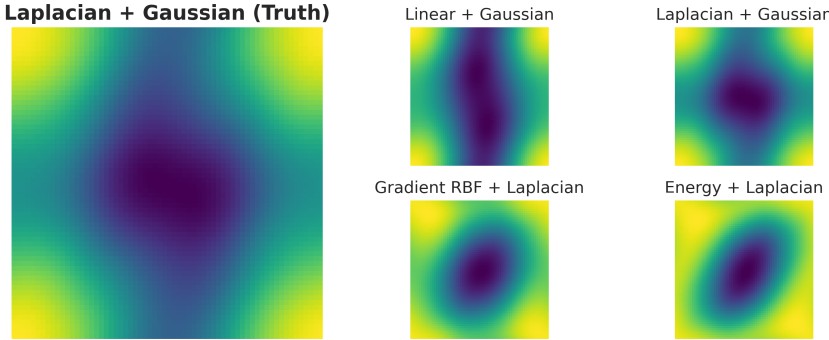

Figure 5: In-context predictions for $(k_x, k_y)$ being (Linear, Gaussian), (Laplacian, Gaussian), (Gradient RBF, Laplace), and (Energy, Laplace) with the data-generating kernel being (Laplacian, Gaussian).

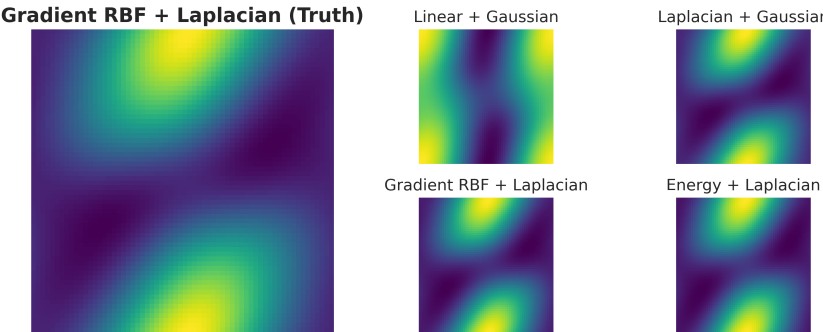

Figure 6: In-context predictions for $(k_x, k_y)$ being (Linear, Gaussian), (Laplacian, Gaussian), (Gradient RBF, Laplace), and (Energy, Laplace) with the data-generating kernel being (Gradient RBF, Laplace).

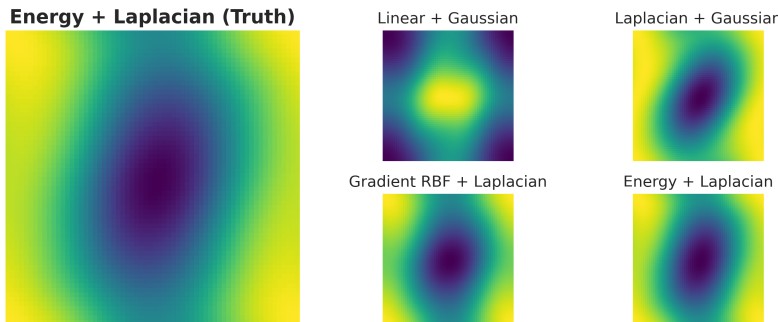

Figure 7: In-context predictions for $(k_x, k_y)$ being (Linear, Gaussian), (Laplacian, Gaussian), (Gradient RBF, Laplace), and (Energy, Laplace) with the data-generating kernel being (Energy, Laplace).

## J  POISSON EQUATION SAMPLES

We provide visualizations of samples obtained by solving the Poisson equation below, as used in the experiments of Section 5.2.1.

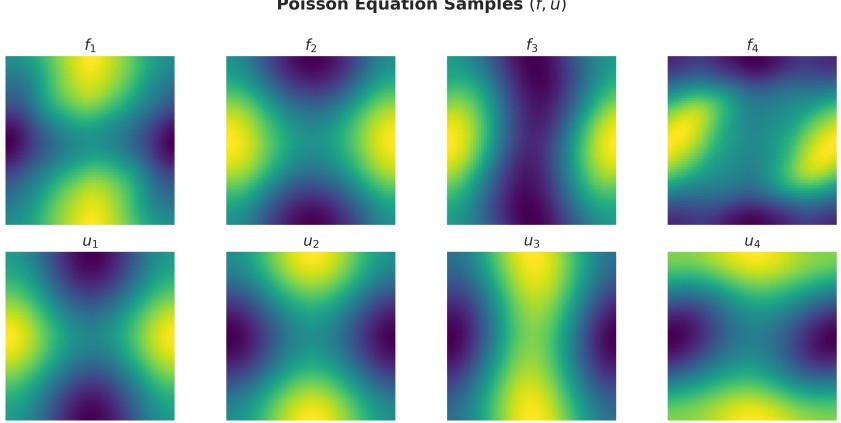

Figure 8: Sample solution pairs generated for the Poisson equation in-context learning task. Samples of $f$ were drawn from a GRF and $u$ solved with an analytic FFT-based Poisson solver.

## K  OPTIMIZATION EXPERIMENT DETAILS

### K.1  OPTIMIZATION KERNEL ASSUMPTIONS

We now provide the proof that Assumption 3.5 holds for the chosen kernel under the particular choice of $k_x$ being Linear and $k_y$ being Gaussian, which follows trivially from properties of the inner product. In particular, for the assumed operator $S : \mathcal{X} \to \mathcal{X}$ with an inverse $S^{-1}$ referenced in Assumption 3.5, notice

$$[\tilde{H}(f^{(1)}, f^{(2)})u] := k_{\text{Linear}}(f^{(1)}, f^{(2)})(k_{\text{Gauss}} * u)$$
$$:= \langle f^{(1)}, f^{(2)} \rangle (k_{\text{Gauss}} * u)$$
$$\implies [\tilde{H}(S^* F_1, S^{-1} F_2)u] = \langle S^* f^{(1)}, S^{-1} f^{(2)} \rangle (k_{\text{Gauss}} * u)$$
$$= \langle f^{(1)}, (S^*)^* S^{-1} f^{(2)} \rangle (k_{\text{Gauss}} * u)$$
$$= \langle f^{(1)}, S S^{-1} f^{(2)} \rangle (k_{\text{Gauss}} * u)$$
$$= \langle f^{(1)}, f^{(2)} \rangle (k_{\text{Gauss}} * u),$$

completing the proof as desired.

### K.2 OPTIMIZATION EXPERIMENT HYPERPARAMETERS

The hyperparameters of the optimization experiment (results presented in Section 5.3) are given in Table 1. The continuum transformer was implemented in PyTorch Paszke et al. (2019). Training a model required roughly 30 minutes to an hour using an Nvidia RTX 2080 Ti GPUs.

Table 1: Hyperparameters used in the continuum transformer training experiment.

| Category | Hyperparameter (Value) |
|---|---|
| Model | Number of Layers: 250
Image Size: $64 \times 64$
$k_x$ $\sigma$: 1.0
$r_\ell$ Initialization: $-0.01$ |
| Dataset | $k_y$ Kernel: `Gaussian`
# In-Context Prompts: 25
# Operator Bases: 30 |
| Training | Optimizer: `Adam`
Learning Rate: 0.001
Epochs: 10
# Samples: 128
Momentum: 0.0 |

## L  NOTATION REFERENCE

We provide below a comprehensive presentation of the notation from the paper.

Table 2: Notation for the data

| Notation | Description |
|---|---|
| $\mathcal{X}$ | Hilbert space in which the input and output functions lie. |
| $f^{(i)}, u^{(i)}$ | $i$-th pair of input and output functions. |
| $Z_\ell$ | A matrix in $\mathcal{X}^{2\times(n+1)}$ whose first row consists of input functions $f^{(i)}; i = 1, \ldots, n+1$, and whose second row consists of output functions $u^{(1)}, \ldots, u^{(n)}, 0$. The zero is a placeholder for the predicted output of $f^{(n+1)}$ which will change as $Z_\ell$ passes through the transformer. |
| $X_\ell$ | The first row of $Z_\ell$. |
| $\mathcal{O}$ | Space of operators in which the true operator lies. This is assumed to be an RKHS. |
| $\mathbb{K}(F)$ | Covariance operator of the output functions $U$ conditioned on the input functions $F$. |

Table 3: Notation for the transformer architecture

| Notation | Description |
|---|---|
| $\mathcal{T}_\ell$ | The $\ell$-th layer of the transformer. |
| $\mathcal{W}_{k,\ell}, \mathcal{W}_{q,\ell}, \mathcal{W}_{v,\ell}$ | Key, query and value operators at the $\ell$-th layer respectively. |
| $\mathcal{M}$ | Mask operator. |
| $\tilde{H}(\cdot, \cdot)$ | Non-linear operator. |
| $r_\ell$ | Scalars parametrizing the value operator at layer $\ell$. |

Table 4: Notation for ICL, gradient descent, and the RKHS framework

| Notation | Description |
|---|---|
| $L(\cdot)$ | In-context loss. |
| $O_\ell$ | Operator at the $\ell$-th gradient descent step. |
| $\kappa$ | Operator-valued kernel which takes an element in $\mathcal{X} \times \mathcal{X}$ to a bounded linear operator on $\mathcal{X}$. |

## M  LLM USAGE

LLMs were not used in the writing of the manuscript; they (specifically Gemini) were used as a tool to help with the debugging of some pieces of the code for the experiments.

