# OpenReview forum: "Continuum Transformers Perform In-Context Learning by Operator Gradient Descent"
_ICLR.cc/2026/Conference — ICLR 2026 Poster_

### Official Review · Reviewer_peRa · 2025-10-28

**Soundness:** 3
**Presentation:** 3
**Contribution:** 3
**Rating:** 6
**Confidence:** 3

**Summary:**

This paper studies in context learning abilities of infinite-dimensional generalizations of the transformer architecture, called continuum transformer and previously introduced by Calvello et al. The authors extend previous finite-dimensional work by Cheng et al. showing that transformers implement gradient descent in-context to infinite dimensional settings for operator learning. The main result is that continuum transformers perform in-context learning by performing gradient descent in an RKHS, under a careful choice of the architecture. The authors conclude by providing numerical experiments on simple operator learning tasks that validate their theoretical findings.

**Strengths:**

1. Despite the technicalities, the paper is well written and rigorous.
2. The topic is timely and relevant, especially as transformer-based architectures are becoming increasingly popular in the field of operator learning with the apparition of foundation models.
3. The main theoretical results are interesting and bring new insights on the in-context learning abilities of transformer architectures in infinite-dimensional settings, and may lead to more practical considerations regarding the choice of the architecture.
4. The numerical experiments, although simple (linear operators), illustrate the theory.

**Weaknesses:**

1. Given the formulation of Thm 3.1, it is quite difficult to interpret the result in practical terms. I am wondering whether, instead of framing it as an existence result for the architecture, it would be more natural to introduce the kernel induced by the architecture. The result also makes a lot of assumptions on the weight matrices. Can the authors comment on these assumptions and whether they imply that one should fix these weights in practice during training?
2. The two assumptions 3.4-3.5 seems quite specific, it would be helpful to have some discussion on how reasonable they are in practice.
3. It seems counter-intuitive to me that one does not need to make any assumption on the operator $\tilde{G}$ that one tries to approximate. Could the authors comment on this?
4. The numerical experiments are performed for a very synthetic task (Laplacian kernel), and would benefit from a more challenging operator learning task (e.g. a time-dependent PDE, or nonlinear problem is the theory extends to nonlinear operators).

**Questions:**

1. In Eq. (1), what measure do you take on the space of functions U?
2. In Eq. (9), shouldn't one consider a relative loss by normalizing by $\|u\|_{n+1}$? Does the analysis extend to this case?
3. I don't think $\mathcal{O}$ is defined before line 209
4. Theorem 3.1: I can't find the definition of $\tilde{H}$ in the main text before Eq. (7). I see it defined in Eq. 17 but it would be helpful to include it in the main text given that it is crucial for the theorem.
5. I guess Prop 3.3 implicitly assumes that the underlying operator is linear so that the conditional distribution of the output is Gaussian? Is it not a strong assumption to assume that the Gaussian kernel is exactly the same as the kernel induced by the architecture?

---

> ### Author Response · Authors · 2025-11-17
>
> We thank the reviewer for their time and questions. We address each of the points successively below.
>
> *Given the formulation of Thm 3.1, it is quite difficult...*
>
> The practical implication is that users should seek to have the kernel employed in the transformer architecture match that of the RKHS the data are drawn from. Alternatively, as the reviewer is suggesting, one can analyze the error induced from having a mismatched kernel even if the true kernel used for the data was different, which is what was empirically done in Section 5.2.
>
> Our results draw from the lines of existing literature that seek to understand finite-dimensional ICL, where matrices are studied that produce inference passes that are equivalent to known algorithms. Even with these results having been proven, however, finite-dimensional transformer weight matrices are nearly never fixed ahead of time in practical applications. This is likely because, while ICL is a desired behavior of networks, it is only one amongst many, another being the recall of patterns observed during the training process. The final weights of deployed transformers, therefore, likely are not those prescribed by these characterizations but instead weights whose behaviors are noisy versions of the idealized behaviors given by these explicit matrices.
>
> *The two assumptions 3.4-3.5 seems quite specific...*
>
> See general response (3) above.
>
> *It seems counter-intuitive to me that one...*
>
> Theorem 3.1 establishes that we *can* perform gradient descent with a continuum transformer: intuitively, the finite-dimensional analog would simply require that the objective upon which we wish to perform gradient decent be sufficiently smooth. In the operator setting, we are implicitly making this assumption by assuming that the operator lies in an RKHS (see Lemma D.4). Notably, Theorem 3.1 does not establish the quality of such an estimator: this is only established in Proposition 3.3, which requires the Gaussian assumption of the conditional distribution.
>
> *The numerical experiments are performed for a very synthetic...*
>
> See general response (2) above.
>
> *In Eq. (1), what measure do you take on the space of functions U?*
>
> In the operator learning literature, we assume there is some data-generating process such that $u_0\sim\mu$ and that the measure of interest is the resulting push-forward measure of this data-generating distribution.
>
> *In Eq. (9), shouldn't one consider a relative loss by normalizing...*
>
> One can consider the relative loss; however, we follow the conventions of the ICL analysis literature, where experiments commonly study the unnormalized error (see [1], [2], and [3]). Gradient descent equivalence for a normalized error may also be exhibited under a different set of weight matrices.
>
> [1] What Can Transformers Learn In-Context? A Case Study of Simple Function Classes
>
> [2] What learning algorithm is in-context learning? Investigations with linear models
>
> [3] Transformers implement functional gradient descent to learn non-linear functions in context
>
> *I don't think $\mathcal{O} is defined before line 209*
>
> The reviewer is correct: we do define it in Appendix K with the rest of the notation but will explicitly introduce it to the main text.
>
> *Theorem 3.1: I can't find the definition of...*
>
> $\widetilde{H}$ is used to denote the nonlinearity when restricted to the $\mathcal{X}$ space to highlight that this acts only on the $\mathcal{X}^{n+1} \times \mathcal{X}^{n+1}$ space rather than the full $\mathcal{Z}^{n+1} \times \mathcal{Z}^{n+1}$ space that $H$ normally acts on. We will mention this more explicitly in the main text.
>
> *I guess Prop 3.3 implicitly assumes that the underlying operator is linear...*
>
> We assume that the conditional distribution of $U|F$ is Gaussian, but the kernel $\kappa$ can be non-linear. Since the activation function $\tilde{H}$ is defined element-wise in terms of $\kappa$, it does not have to be linear.
>
> Regarding kernel misalignment, classical analysis from the functional data analysis literature assumes that the data or the parameter to be estimated lies in an RKHS. For instance, [6] showed that kernel alignment is necessary for functional linear regression and proposed a data-adaptive method to achieve this. However, we see empirically in Figure 1 that the estimator error generally decays even under mismatches, suggesting that a stronger theoretical result exists depending on some “similarity” measure between kernels. Our method of analysis (see Section F.2) involves demonstrating a geometric decay of error under the $(I-\delta \hat{\mathbb{K}})$ operator; an extension of this proof strategy, where $\hat{\mathbb{K}}$ is replaced by a perturbed operator, would likely enable a generalized result to be established under kernel mismatch.
>
> [6] Cai, T. T. and Yuan, M. ‘Minimax and adaptive prediction for functional linear
> regression’, Journal of the American Statistical Association, 2012.

---

> > ### Comment · Reviewer_peRa · 2025-11-26
> > **Response to the authors**
> >
> > I thank the authors for the detailed response which addressed most of my comments regarding the theory. I have also read other reviewers' comments and, while I agree that this work is mostly theoretical, I believe validating the theory on more realistic tasks (such as an operator learning benchmark) would help the paper. For this reason, I will keep my original score and appreciation of the paper, which remains positive.

---

> > > ### Author Response · Authors · 2025-11-28
> > >
> > > We thank the reviewer for reading through the rebuttals. In response to the raised concerns about the behavior of the method under the setting of a PDE operator learning task, we have introduced a new experiment to the manuscript (in Section 5.2.1): please see the recently posted general comment for details.

---

### Official Review · Reviewer_iXYP · 2025-10-28

**Soundness:** 3
**Presentation:** 2
**Contribution:** 3
**Rating:** 6
**Confidence:** 3

**Summary:**

This paper presents a rigorous theoretical analysis of continuum transformers, the function-space analog of standard transformers, when applied to operator learning problems, such as those arising in PDE modeling. The authors show that:

- If the query, key, and value operators are chosen appropriately, each layer of a continuum transformer performs one exact step of gradient descent in an operator-valued RKHS

- As the number of layers tends to infinity, the transformer’s output converges to the Best Linear Unbiased Predictor, equivalent to the Bayes-optimal estimator of a Hilbert-space Gaussian process.

- When the model is trained end-to-end, its parameters naturally converge to the same fixed-point configuration that realizes the gradient-descent dynamics

Empirical studies are limited but targeted:

- The authors validate the gradient-descent interpretation when the attention kernel matches the data-generating kernel.

- They train a 250-layer continuum transformer on synthetic Gaussian-process data to verify the convergence of learned query/key operators toward the predicted fixed-point structure.

The paper is primarily theoretical but provides convincing toy experiments supporting the analytical results.

**Strengths:**

- Originality: Extends the “transformer = gradient descent” theory from vector spaces to operator-valued RKHSs, a nontrivial and conceptually elegant generalization.

- The mathematical treatment is rigorous and self-contained, including a generalized Representer Theorem for operator-valued kernels.

- The results provide a clear mechanistic interpretation of in-context learning for continuum transformers.

- Toy experiments (BLUP matching and operator-alignment studies) convincingly illustrate the theorems.

- The work strengthens the theoretical foundation of operator learning and may guide better architectures for PDEs

**Weaknesses:**

- Lack of practical tokenization discussion.

It remains unclear how tokenization is defined in PDE-learning settings. While a “next-token prediction” formulation is appropriate for fixed time-stepping tasks (e.g., 6-hour weather forecasts), such an in-context learning paradigm does not naturally extend to continuous-in-time scenarios where predictions are required at arbitrary temporal resolutions. Moreover, most PDE and computer-vision problems require some form of spatial tokenization to capture local correlations across the domain (e.g., patchifying a sample function). In this functional context, the notion of “next-patch prediction” becomes conceptually ambiguous and difficult to formalize.

- Limited empirical scope.

All experiments are synthetic. The theory’s applicability to real PDE problems (e.g., weather trajectory generation or high-resolution dynamics) is not demonstrated.

- Scalability concerns.

It is unclear how the proposed theory would extend to realistic, large-scale problems, such as generating full weather trajectories. In such settings, deploying a 250-layer network (as used in Section 5.3) would be computationally impractical. It would therefore be valuable to analyze how the average similarity score evolves as the model depth increases, from shallow configurations (1–10 layers) through intermediate (10–50) to deep regimes (50–250). My expectation is that the network would not reach the exact fixed point predicted by Theorem 3.6, but would likely approach it asymptotically.

- Dense notation.

The presentation could benefit from more intuition and conceptual figures summarizing main theorems and propositions.

- Lack of clarity on computational and memory complexity.

The continuum formulation replaces matrix multiplications with integral operators, but the paper does not discuss the resulting computational or memory scaling. It is unclear whether the operator-valued attention can be implemented efficiently in discretized settings or whether it incurs quadratic cost in the number of spatial degrees of freedom. A short discussion comparing computational requirements with standard transformer or neural-operator architectures would strengthen the paper’s practical relevance.

**Questions:**

- How do you envision “tokens” being defined in PDE settings? Are they spatial patches, basis coefficients, or entire functional evaluations?

- Could your analysis extend to models queried at arbitrary times (e.g., continuous-in-time diffusion or flow prediction)?
	​
- Would the cosine-similarity convergence in §5.3 persist for shallower models? A quantitative study varying the number of layers could clarify practical limits.

- Do you expect the same theoretical behavior when training on PDE datasets like Navier–Stokes or Darcy flow, where discretization and boundary conditions break some of the theoretical assumptions?

**Details Of Ethics Concerns:**

/

---

> ### Author Response · Authors · 2025-11-17
>
> We thank the reviewer for their time and questions. We address each of the points successively below.
>
> *Lack of practical tokenization discussion.*
>
> Our analysis, and the in-context learning of continuum transformers more generally, is independent of what the $(f, u)$ pairs represent. The most common setting is that mentioned by the reviewer, namely discretized unrolling of a spatiotemporal PDE; the neural operator literature is dominated by this setting, where the continuity in the spatial domain is considered with temporal dynamics discretized over fixed increments. The tokens here are then functions at various time points.
>
> This, however, is not the only application: one could also have the tokens $(f_ {\Omega_i}, u_ {\Omega_i})$ be the functions restricted to patches of the domain $\Omega_i$, although there are practical issues with this due to the dependency between patches from boundary conditions. If, however, one is able to circumvent this issue, the results presented here immediately extend to that setting.
>
> *Limited empirical scope.*
>
> See general response (2) above.
>
> *Scalability concerns.*
>
> See general response (1) above.
>
> *Dense notation.*
>
> We tried to streamline the notational burden by following standard notational conventions wherever available and providing a summary in Appendix K.
>
> *Lack of clarity on computational and memory complexity.*
>
> We note that, as discussed above, the experiments were intended to demonstrate the alignment between the theoretical characterization of the in-context learning mechanism more than suggesting an approach for using ICL in practical settings, for which reason we did not report the computational cost of the experiments. For completeness, however, we report the numbers below for the wall-clock times of inference as a function of the number of layers (i.e. the cost of conducting a single trial of Experiment 1 from Section 4.2). As expected, the cost simply grows linearly with the number of layers.
>
> | # Layers | Time (s) |
> | :--- | :--- |
> | 100 | 0.80 |
> | 250 | 1.52 |
> | 500 | 2.81 |
> | 750 | 3.76 |
> | 1,000 | 5.07 |
>
> *How do you envision tokens being defined...*
>
> See response to the first question above.
>
> *Could your analysis extend to models...*
>
> While we mentioned that the most common application of operator ICL is for discretized spatiotemporal PDE rollout, so long as the operator for a given ICL sample is fixed, our analysis results immediately apply. This means, for any given sample, the rollout pairs have to have a fixed time increment, i.e. the tokens have the form $u_ {i}^{(0)}, u_ {i}^{(\Delta T)},...,u_ {i}^{((T-1)\Delta T)}$ and cannot be, for instance, $u_ {i}^{(0)}, u_ {i}^{(\Delta T)},u_ {i}^{(4 \Delta T)}$, since then the operator is mapping from token 1 to 2 differs from that mapping from 2 to 3. In this latter case, the very notion of doing ICL operator learning is ill-framed, since there is *no* single operator of interest to learn.
>
> However, the increments *across* samples *can* differ, i.e. sample $i$ can have increments $\Delta T$ and sample $j$ have $\Delta T’$ so long as the resulting operators all lie in a common RKHS. An interesting question for further study is establishing what conditions ensure that these operators of differing increments lie in a fixed RKHS.
>
> *Would the cosine-similarity convergence...*
>
> The results established in Theorem 3.6 are not asymptotic results, meaning this fixed-point result is independent of the depth of the network. The results of Section 5.3, however, demonstrate convergence to this particular fixed point, which is stronger than establishing that it *is* a fixed point. This behavior, however, should be even more robust in shallower networks, where the lower dimensionality of the optimization landscape reduces the number of alternate fixed points the transformer may otherwise converge to. We will conduct this additional experiment to add to the final supplement of the paper.
>
> *Do you expect the same theoretical...*
>
> There are two main structural assumptions upon which our results stand. The first is how robustly the ICL parameters are recovered with gradient descent under violations of the structural assumptions on the kernel. Even for the kernels where the symmetry does not necessarily hold (i.e. all kernels other than the linear one in the experiments), we see in Figure 2 that the empirical results point to the consistency of the convergence guarantees.
>
> The second is that the operators all lie in an RKHS. In particular, this amounts to studying, for a parameterized PDE, such as $u = D_ {a} f$ for $x\in\Omega$ and similarly over the boundary, under what distributional assumption on the parameters $a$ do the solution operators $\mathcal{O}$ lie in an RKHS? Given that we have established ties between operator gradient descent and transformer ICL in the general RKHS setting, this work suggests that such characterization is likely a fruitful path for further inquiry.

---

> > ### Comment · Reviewer_iXYP · 2025-11-26
> >
> > I thank the authors for their clarifications and the additional effort in the rebuttal. While I understand that the focus of the paper is predominantly theoretical, the absence of experiments on real PDE datasets leaves the practical relevance of the findings somewhat unclear and uncomplete. I believe that including even a small number of PDE benchmarks would substantially strengthen the paper and provide a more complete picture of the studied mechanisms in action. Given the points above and despite the helpful clarifications, I intend to maintain my current score.

---

> > > ### Author Response · Authors · 2025-11-28
> > >
> > > We thank the reviewer for reading through the rebuttals. In response to the raised concerns about the behavior of the method under the setting of a PDE operator learning task, we have introduced a new experiment to the manuscript (in Section 5.2.1): please see the recently posted general comment for details.

---

### Official Review · Reviewer_sPh8 · 2025-10-29

**Soundness:** 3
**Presentation:** 3
**Contribution:** 2
**Rating:** 6
**Confidence:** 4

**Summary:**

The paper lifts the in-context learning approach from finite dimensional transformers to continuum transformers acting on function spaces. Replacing softmax with a PSD operator valued kernel, each layer implements a gradient descent step for the in-context least squares objective in an Operator Valued Reproducing Kernel Hilbert space (OVRKHS) and depth corresponds to more steps. Under a $\kappa$-Gaussian distribution and matched $\kappa$ attention kernel, in the infinite depth limit the method equals a Bayes optimal predictor (BLUP/kriging). A pre-training analysis, under strong assumptions, shows that whitening Q/K and considering a residual injecting V forms  stationary solutions, suggesting that the model can learn parameters in a GD manner. The experiments validate the theory for a simple test.

**Strengths:**

- The paper presents a non-trivial extension of in context leaning to an operator valued, continuous, setting. The math is elegant have technical substance and can be used for further analysis.

- Considering operator valued kernels for this problem and their representer theorem is elegant and makes defining each layer as one gradient descent step clean. The analysis for the fixed point pre training and the connection to BLUP is nice despite the strong assumptions.

- The attention, the values and how each part of the method is constructed is clearly stated.

Overall, I find this paper to be mathematically elegant and insightful.

**Weaknesses:**

- The Bayes optimality/BLUP limit require the data distribution to be $\kappa$-Gaussian and also an exact kernel match. In practice, the input functions come from non Gaussian distributions and the kernels might be misspecified.

-  Rotational symmetry and equivariance are strong assumptions. Correct me if I am wrong but I believe that a global invariance in whitened coordinates might not hold due to anisotropies or boundary conditions.

- All the proofs are performed in the function space, however the implementations in practice are discretised. Maybe there should be an analysis linking the continuum to the discrete implementation and what errors this induces.

- Operator valued kernels can be expensive. The paper bypasses that using a separable form trick, scalar similarity * cheap operator, and tiny prompts however a discussion should be presented on estimating rich kernels and larger contexts.

- The theory requires a PSD and symmetric kernel which standard softmax is not. So, what the authors use is not technically an attention kernel because it is not row normalized. This is not that clear. I suggest to the authors to use the kernel construction considered in “Learning Operators with Coupled Attention” where they consider a PSD/symmetric/universal kernel that is also normalized. The authors there consider a scalar kernel I believe that this can be adapted to your method in a straightforward manner (see Micchelli “On learning vector-valued functions” and Caponnetto “Universal Multi-task Kernels”).

 - The empirical validation is very limited. You consider no PDE benchmarks, no baselines, ICON or FNO and a meta-learner, no ablations for kernel mismatch or symmetry braking.

**Questions:**

- How is the projection error bounded when considering a truncated spectral basis?

- Is there a way to relax the whitening assumption by maybe considering a spectral normalisation of W_q, W_k and an adaptive step size?

- How sensitive is the fixed point analysis to broken symmetries from non-periodic BCs, irregular meshes etc?

- Please add at least one PDE example and compare to an FNO+meta learning or ICON approach.

---

> ### Author Response · Authors · 2025-11-17
>
> We thank the reviewer for their time and questions. We address each of the points successively below.
>
> *The Bayes optimality/BLUP limit require the data distribution to be…*
>
> Optimal prediction for non-Gaussian data is an interesting field of research in spatial statistics. The kriging predictor may not be the optimal predictor for unsampled locations for a non-Gaussian process. This is a challenging problem, and recently [4] proposed a copula-based model and a semiparametric estimator that works well with non-Gaussian data. However, it is not quite clear how this method can be readily adapted to neural architectures.
>
> Regarding kernel misalignment, classical analysis from the functional data analysis literature assumes that the data or the parameter to be estimated lies in an RKHS. For instance, [6] showed that kernel alignment is necessary for functional linear regression and proposed a data-adaptive method to achieve this. However, we see empirically in Figure 1 that the estimator error generally decays even under mismatches, suggesting that a stronger theoretical result exists depending on some “similarity” measure between kernels. Our method of analysis (see Section F.2) involves demonstrating a geometric decay of error under the $(I-\delta \hat{\mathbb{K}})$ operator; an extension of this proof strategy, where $\hat{\mathbb{K}}$ is replaced by a perturbed operator, would likely enable a generalized result to be established under kernel mismatch.
>
> [4] Agarwal et al. ‘Copula-based multiple indicator kriging for non-Gaussian random fields’. Spatial Statistics, 2021.
>
> [6] Cai, T. T. and Yuan, M. ‘Minimax and adaptive prediction for functional linear
> regression’, Journal of the American Statistical Association, 2012.
>
> *Rotational symmetry and equivariance are strong assumptions…*
>
> See general response (3) above.
>
> *All the proofs are performed in the function space, however…*
>
> We agree that the finite-projection approach is likely an alternate proof strategy that would work. We believe, however, that the operator approach is a more elegant framework to contribute to the community. Now that we have gone through this formalism, the broader community can **directly use these analysis strategies** to further study in-context learning or perhaps for completely unrelated inquiries, which is highly valuable, since it means theoreticians can, in many ways, simply follow their finite-dimensional intuitions and defer to our infinite-dimensional results to rigorously justify their results. In other words, our proof strategies suggest to other researchers working on similar problems that they need not be bogged down in the details of the error convergence minutiae that often arise in approaches relying on finite projections and can instead work with these clean abstractions.
>
> Similarly, this work opens up the space for who can contribute to further the theoretical study of operator ICL. Much of the classical optimization community, for instance, may not be intimately familiar with the mathematical formalisms required around operators. However, with our provided formalism, they can provide insights with little change from how they would approach finite-dimensional analyses.
>
> This, in fact, is a question more broadly applicable to the functional data analysis community: in many FDA cases, one can argue that similarly using finite-dimensional projection would suffice in place of turning to infinite-dimensional functions. However, the benefits there are similar: functions are clean mathematical objects for use in analysis.
>
> *The theory requires a PSD and symmetric kernel which standard softmax is not…*
>
> Indeed, as the reviewer points out, since we require the non-linearity to match the kernel of the RKHS or that of the Gaussian process, softmax cannot be employed in the current framework. However, if we choose an exponential kernel, it gives the same results as softmax up to a scaling factor, from which our results immediately extend to this common parameterization.
>
> The reviewer pointed us in the direction of the paper on “Learning Operators with Coupled Attention” which employs a softmax attention coupled with an integral transform. However, the architecture they propose requires conversion of input functions to a discrete set (feature encoding). This step is contrary to the aim of using continuum transformers, which retain the infinite-dimensional nature of the data in theory to accurately perform algorithms in the operator space. If the feature encoding step is removed, then it is unclear what the infinite-dimensional counterpart to softmax is.
>
> *The empirical validation is very limited…*
>
> See general response (2) above.

---

> > ### Author Response · Authors · 2025-11-17
> >
> > *How is the projection error bounded when considering a truncated spectral basis?*
> >
> > As mentioned above, this is likely a feasible analysis approach. We believe that the analysis would follow a similar technique as used in [1]; in this work, however, we need the additional assumption that the functions are drawn from a smooth space, such as the Sobolev space $\mathcal{H}^{s}$, whereby the unseen spectral errors can be bounded. In particular, under specific parameters of the RKHS kernel, such smoothness can be induced over the function space to arrive at the desired conclusions. However, as stated above, we wished to retain the functional viewpoint to more elegantly arrive at these results.
> >
> > [1] Operator Learning for Schrödinger Equation: Unitarity, Error Bounds, and Time Generalization
> >
> > *Operator-valued kernels can be expensive. The paper bypasses that using a separable form trick…*
> >
> > As mentioned in the paper, these kernels were chosen by virtue of being typical of the RKHS literature in functional data analysis. However, as we expand to exploring this ICL phenomenon in PDE settings, the kernels may likely not be exactly expressible in this decomposed form, which suggests that an interesting direction to explore would be error characterization under kernels *approximations* that are decomposable.
> >
> > [2] Operator-valued kernels for learning from functional response data
> >
> > *Is there a way to relax the whitening assumption…*
> >
> > Theorem 3.6 establishes the fixed-point nature of $\mathcal{W}_ {k,\ell}$ and $\mathcal{W}_ {q,\ell}$ when restricted to the manifold $\mathcal{S}$. As the reviewer is suggesting, this indicates that, if such operators were normalized to remain on this manifold, a similar result could be established by allowing for an adaptive step size to play the role of $b_ {\ell}$ and $c_ {\ell}$. This, however, would still require the step adaptation to, in some way, relate to the distributional properties of $F$. Intuitively, the data-generating distribution of the inputs must manifest in some parameter of the optimization analysis, although the authors agree that there are likely some ways of more directly relating these parameters than the mathematically clean treatment afforded by the whitened analysis.
> >
> > *Please add at least one PDE example and compare to…*
> >
> > See general response (2) above.
> >
> > *How sensitive is the fixed point analysis to broken symmetries from non-periodic BCs, irregular meshes etc?*
> >
> > As mentioned in the Discussion section of the paper, the PDE setting was largely a backdrop to the contributions of the paper. Given that we have now established these general results, however, it motivates the study in settings common in PDE ICL. In particular, this amounts to studying, for a parameterized PDE, such as $u = D_ {a} f$ for $x\in\Omega$ and similarly over the boundary, under what distributional assumption on the parameters $a$ do the solution operators $\mathcal{O}$ lie in an RKHS? We are currently investigating this as an extension to the current work.

---

> ### Comment · Reviewer_sPh8 · 2025-11-28
>
> Thank you for the detailed review. Some of your statements are not complete, but I accept them.
>
> In my understanding, the "attention" choice and definition should be explicitly discussed in the paper, because it matters a lot to the analogies you are making here.
>
> Your method relies on a symmetric, positive-semidefinite operator values kernel which is not an attention kernel in the commonly used sense. Standard softmax attention does not have this property because the row-wise, data-dependent normalization breaks symmetry and positive semidefiniteness. I believe that making this explicit would clear any confusion about what you mean by attention here, a kernel averaging  instead of a row stochastic map.
>
> Overall, I find your paper elegant, but I still believe that you need to have experiments to show that it is also applicable in realistic scenarios.  The VICON paper reference you provide considers a softmax attention, multiple heads and a few layers, and other architecture choices which are either not supported by your theory or not clear how your method behaves under this setup, so I do not see the analogy here. Moreover, I am not sure about the trade-off between computational cost and   expressiveness of the kernel approximation because as we discussed operator valued kernels are generally very expensive and need to be approximated. All of these would be made clear if you had an experiment.
>
> For this reason, even though I find your method to be elegant and your paper nice, I believe that it needs work to practically show your claims and stress test your method under realistic assumptions. I believe that this will raise the potential impact or your method.
>
> P.S. My suggestion was to consider the kernel in "Learning Operators with Coupled Attention" inside the softmax which has the properties you need and it is also normalized.

---

> > ### Author Response · Authors · 2025-11-28
> >
> > We thank the reviewer for reading through the rebuttals. In response to the raised concerns about the behavior of the method under the setting of a PDE operator learning task, we have introduced a new experiment to the manuscript (in Section 5.2.1): please see the recently posted general comment for details.

---

### Official Review · Reviewer_uu2i · 2025-11-01

**Soundness:** 4
**Presentation:** 4
**Contribution:** 3
**Rating:** 8
**Confidence:** 4

**Summary:**

This work extends the line of work showing that Transformers perform gradient descent in-context during inference to continuum Transformers (defined on function spaces) and operator learning. The main theoretical contributions are that continuum Transformers can implement operator gradient descent in a reproducing kernel Hilbert space (RKHS) of operators and that in the infinite-depth limit, the model converges to the Bayes-optimal (kriging) predictor. The authors prove this by providing a weight construction implementing operator gradient descent. Furthermore, they prove that their choice of parameters also represents a stationary point of the training loss under gradient flow over the space of functionals. They empirically validate their theoretical claims on operator regression tasks.

**Strengths:**

- The paper is novel and investigates an important question, probing the fundamentals of continuum Transformers for in-context operator learning. This is a problem that may be of interest to folks in the in-context learning theory and operator learning/PDE foundation model communities.
- The paper is well-written and clear. The theoretical results are interesting and well done, and the authors validate their theoretical results empirically.

**Weaknesses:**

As far as I understand, the work does not provide any analysis of how many functional gradient descent steps / number of continuum Transformer layers it would take to achieve convergence on the in-context operator learning problem. In my opinion, the work as it stands already provides enough results to recommend acceptance, but I would be curious to know if such results are possible (see questions).

Minor:
- The authors could more clearly emphasize the significance and limitation of the theoretical claim in Section 3.4 / Theorem 3.6. As stated, the theorem establishes that the proposed parameter configuration constitutes a stationary point (fixed point) of the training dynamics in the continuum limit, but it does not imply that this configuration is the unique or global minimizer of the loss, nor that it corresponds to the explicit solution of the gradient flow. I understand this type of result is standard in recent in-context learning theory (and I believe the result itself is valuable to the community), highlighting it explicitly would help readers interpret the scope of the claim.
For example, lines 61 and 273 could note that the parameters form a critical point, not necessarily a global minimum.
- I think the writing would be slightly clearer if the authors allocated a specific subsection in the paper for discussion of the novelty/importance of the proof techniques, rather than scattering such comments throughout.

**Questions:**

- As far as I understand, the theoretical results characterize the continuum transformer’s behavior in the infinite-depth setting, but do not provide any results or intuition about the rate of convergence. Would it be possible to extend the authors' analysis to provide convergence rate results (e.g. how many steps/layers are required for the in-context operator to approach the Bayes-optimal predictor)? Or alternatively, have the authors observed any empirical scaling trends relating depth to approximation quality?
- In Figure 2, the authors attempt to provide empirical validation of their fixed-point claim by evaluating the cosine similarity of the key/query operators across layers during training as a proxy. I'm wondering if it would be possible to use a similar proxy to visualize the rate of convergence of the in-context operator prediction as the model evaluates layer-by-layer? For example, by evaluating the cosine similarity or relative error between the predicted operator at layer $\ell$ and at layer $\ell-1$, or between the layer $\ell$ prediction and the Bayes-optimal operator? Such a diagnostic might more directly visualize the model prediction's evolution under the proposed functional GD.
- Is it possible to extend this analysis of infinite-dimensional prefix-ICL to the infinite-dimensional causal-ICL setting, where given $n$ ICL examples, the model is supervised on all $n-1$ subproblems (given $i-1$ examples, predict the $i$th, for all $i \leq n$) in parallel? (e.g. as in [1])

1. Bai et al., 2023. Transformers as Statisticians: Provable In-Context Learning with In-Context Algorithm Selection

---

> ### Author Response · Authors · 2025-11-17
>
> We thank the reviewer for their time and questions. We address each of the points successively below.
>
> *The authors could more clearly emphasize the significance and limitation of the theoretical claim in Section 3.4 / Theorem 3.6...*
>
> Yes, as the reviewer correctly points out, this result establishes that such a point is a critical point of the optimization landscape. We agree that this has limitations in its scope of conclusions. However, we believe this is a key first step in the analysis of operator in-context learning, given that no works have theoretically investigated this phenomenon. We believe this initial groundwork now motivates further investigations into whether stronger claims on global convergence are true and, if so, can be similarly proven.
>
> *I think the writing would be slightly clearer...*
>
> We believe that keeping the discussion of the techniques close to the statement of the theorem itself makes it more immediately clear why such techniques are useful for proving the individual results, since the result is still fresh in the reader’s mind. That said, with the additional space provided in the camera-ready submission, we agree it would be worthwhile adding a preface roughly describing what techniques were developed over the paper and then point them out more granularly throughout.
>
> *As far as I understand, the theoretical results characterize...*
>
> To clarify, the result showing the equivalence of ICL to operator gradient descent (Theorem 3.1) is a non-asymptotic result. However, the reviewer is correct in stating that the result of recovering the BLUP is an asymptotic result. The convergence rate, however, immediately follows from slightly modifying the final steps of the proof, where we relied on the geometric convergence of the sum to demonstrate equality in the limit. If we instead consider the finite-layer intermediate values, we see that the errors behave as follows.
>
> As mentioned in the proof, the BLUP is given by $u^{(\*)} = \nu^T \hat{\mathbb{K}}^{-1}\hat{U}$. Additionally, by the final line of the proof, we have that $u^{(n+1)}_ {\ell+1} = - \delta \nu^T \sum_ {k=0}^{\ell}(I-\delta \hat{\mathbb{K}})^k \hat{U}$. Instead of considering the infinite limit of this series, we consider the finite-layer value and analyze the error:
>
> \begin{align*}
>     u^{(n+1)}_ {\ell+1}
>     &= - \delta \nu^T \sum_ {k=0}^{\ell}(I-\delta \hat{\mathbb{K}})^k \hat{U} \\\\
>     &= - \delta \nu^T (I-(I-\delta \hat{\mathbb{K}})^{\ell})(I-(I-\delta \hat{\mathbb{K}}))^{-1} \hat{U} \\\\
>     &= -\nu^T (I-(I-\delta \hat{\mathbb{K}})^{\ell})\mathbb{K}^{-1} \hat{U}
> \end{align*}
>
> Recalling that our ICL prediction is given by the negated value ("We predict $u^{(n+1)}$ as $-[\mathcal{T}(z)]_ {2,n+1}$"), this means the error at layer $\ell+1$ is given by
>
> \begin{align*}
>     \| u^{(n+1)}_ {\ell+1} - u^{(\*)} \|_ {\mathrm{op}}
>     &= \left\| \left( \nu^T (I-(I-\delta \hat{\mathbb{K}})^{\ell})\mathbb{K}^{-1} \hat{U} \right) - \nu^T \hat{\mathbb{K}}^{-1}\hat{U} \right\|_ {\mathrm{op}} \\\\
>     &= \left\| \nu^T (I-\delta \hat{\mathbb{K}})^{\ell}\mathbb{K}^{-1} \hat{U} \right\|_ {\mathrm{op}} \\\\
>     &\le \| \nu^T \|_ {\mathrm{op}} \| (I-\delta \hat{\mathbb{K}})^{\ell} \|_ {\mathrm{op}} \| \mathbb{K}^{-1} \|_ {\mathrm{op}} \hat{U} \|_ {\mathrm{op}}
> \end{align*}
>
> By construction, we had that $\rho := \|I-\delta \hat{\mathbb{K}}\|_ {\mathrm{op}} <1$. Therefore, this error decays geometrically with a factor $\rho < 1$.
>
> *In Figure 2, the authors attempt to provide empirical validation...*
>
> See general response (1) above.
>
> *Is it possible to extend this analysis of infinite-dimensional prefix-ICL…*
>
> Given that Theorem 3.1 and Proposition 3.3 are independent of the training procedure, as they focus on the explicitly exhibited operators under which there is equivalence to gradient descent, these results remain equivalent in the causal and prefix cases. The prefix change, therefore, only manifests in the results of Section 3.4, where we demonstrate the fixed-point nature of the exhibited operators. However, given that the indexing of *n* (i.e., at which position the in-context prediction is being made) is arbitrary in the proof, this proof immediately extends to the causal setting. In other words, since the causal setting considers each of the *n-1* subproblems in parallel, you can equivalently view this as considering *n-1* separate "training examples" for each of the original samples; the fact that they are being computed in parallel is just an implementation detail for speedup. In turn, given that *n* was arbitrary as stated above, the argument of the proof similarly holds in the causal setting.

---

### Author Response · Authors · 2025-11-17

We thank all the reviewers for their time and questions. We address some general concerns raised by reviewers here and address individual concerns in the responses below.

*Updated experiments*

We have updated Section 5.2 and Figure 1 in the attached writeup to add in the prediction of the explicitly constructed BLUP as a reference point in the figure. This demonstrates how, over the layers $\ell$, the predictions converge to those of the BLUP and that the limiting behavior matches that exhibited by the theory.

*Limited scope of comparisons in experiments*

Some reviewers mentioned the scope of comparisons in experiments. We would like to emphasize that the core contribution of our paper was a **theoretical characterization** of an empirically observed phenomenon. Previous works, such as [1], [2], and [3], have provided extensive empirical evidence that shows that continuum transformers are capable of performing in-context learning. Therefore, this architecture **already** has been demonstrated to perform in-context learning empirically for PDEs, meaning the further recapitulation of this point was not of interest in our experiments.

We, thus, sought **not** to extend these empirical findings in our work but rather to provide an **explanation** for these behaviors. While the most common application of such abilities is for meta-learning PDE operators, this was largely a motivator for a more general characterization of this ability, analogous to previous works that have used empirically observed in-context learning abilities of transformers from the context of LLMs to study more abstractly a mathematical characterization of transformers separate from this setting.

This is why our experiments were focused on validating that the in-context learning behavior seems to emerge through the proposed gradient descent mechanism we characterized, instead of focusing on comparisons such as those to alternate meta-learning strategies, such as FNO+meta learning or ICON approaches.

[1] VICON: Vision In-Context Operator Networks for Multi-Physics Fluid Dynamics Prediction

[2] Zebra: In-Context Generative Pretraining for Solving Parametric PDEs

[3] In-Context Operator Learning for Linear Propagator Models

*Generality of Assumptions 3.4 and 3.5*

As discussed in Section 5.3, we note these properties hold for commonly encountered operator kernels, such as that considered in the experiment, as discussed in Appendix G.1. In addition, even for the kernels where this symmetry does not necessarily hold (i.e. all kernels other than the linear one), we see in Figure 2 that the empirical results point to the consistency of the convergence guarantees. As mentioned in the final line of the Discussion, this, therefore, points to there being a stronger version of this statement where those assumptions are relaxed, given this consistency of findings.

In Appendix G.1, it was demonstrated to hold for $\kappa$ defined with $k_{x}$ being linear and $k_{y}$ being Gaussian. However, this property holds for any choice of $k_y$ with $k_x$ taken to be the linear kernel, where the proof follows identically, i.e.

\begin{align*}
    [\tilde{H}(f^{(1)}, f^{(2)})u]
    &:= k_{\mathrm{Linear}}(f^{(1)}, f^{(2)}) (k_y * u) \\\\
    &:= \langle f^{(1)}, f^{(2)} \rangle (k_y * u) \\\\
    \implies
    [\tilde{H}(S^{\*} F_1, S^{-1}F_2)u]
    &= \langle S^{\*} f^{(1)}, S^{-1}f^{(2)} \rangle (k_y * u) \\\\
    &= \langle f^{(1)}, (S^{\*})^{\*} S^{-1}f^{(2)} \rangle (k_y * u) \\\\
    &= \langle f^{(1)}, S S^{-1}f^{(2)} \rangle (k_y * u) \\\\
    &= \langle f^{(1)}, f^{(2)} \rangle (k_y * u),
\end{align*}

---

> ### Author Response · Authors · 2025-11-28
>
> **New Experiment:** *In-Context Estimation for PDE Task*
>
> We have added a new experiment to the manuscript as Section 5.2.1 that explores the ICL estimation hypothesis for a practical PDE learning task. In this setup, we are solving the 2D Poisson Equation, i.e. $u = \nabla f$, seeking to estimate the solution operator $\mathcal{G} : f \to u$ from a dataset $\mathcal{D} := \\{ (f_i, u_i) \\}$. Unlike in the experiments of Section 5.2, we do **not** a priori know the kernel of the RKHS from which the data are drawn, as it is not explicitly constructed; instead, the data are generated by sampling $f_i\sim\mathrm{GRF}(\alpha,\beta,\gamma)$ and then solving the Poisson equation to obtain the corresponding $u_i$. We, therefore, were interested in seeing if the same phenomena exhibited in the experiment of Section 5.2 (of estimation ability improving with depth from taking more operator gradient descent steps in-context), would extend to this setting, where the architecture kernel likely does not match that of the underlying RKHS.
>
> We see from the experiment results shown in Section 5.2.1 that this behavior is robust and continues to be exhibited in this PDE learning setting. Greater details about both the setup and discussion are given in the updated writeup.

---

### Meta-Review · Area_Chair_J5rU · 2025-12-28

**Summary:**

The paper’s main result is a rigorous analysis to explain the prior observation that continuum transformers perform in-context operator learning. The paper establishes an equivalence between the in-context learning of continuum transform in the forward pass and the gradient descent in an operator RKHS, under a careful choice of model architecture. Moreover, the authors prove that their choice of parameters represent a critical point of the training loss. The paper empirically validates the theoretical claim on simple operator learning tasks.

The reviewers shared positive feedbacks on the novelty and importance for addressing the question whether “transformer = gradient descent” for in-context learning during inference in operator-valued RKHS, which is a non-trivial generalization from finite-dimension vector spaces in the prior work.

The reviewers also raised several concerns about the interpretability of tokenization in continuous time, connections with the classical softmax attention kernel, and the limited empirical studies in the PDE setting. In the rebuttal, the authors added a Poisson equation numeric example in addition to the synthetic tasks, which helped to clarify the relevance of the derived theory to realistic scenarios.

**Reviewer Concerns:**

Based on the discussions, there appears to have a converging consensus that this paper is largely of theoretical contribution to understanding the in-context learning of transformer architectures in the operator learning setting. More empirical validation is likely to increase the impact of the paper.

**Reviewer Scores:**

4 reviewers submitted their comments and scores (6/6/6/8) with confidence (3/4/3/4), with average score 6.5 and average confidence 3.5.

I concur with the reviewers and think the initial positive scores should be maintained.

---

### Decision · Program_Chairs · 2026-01-26

Accept (Poster)